# Evaluating a landscape-scale daily water balance model to support spatially continuous representation of flow intermittency throughout stream networks

Songyan Yu[1], Hong Xuan Do[2,3,4], Albert I. J. M. van Dijk[5], Nick R. Bond[6], Peirong Lin[7], Mark J. Kennard[1]

[1]Australian Rivers Institute and School of Environment and Science, Griffith University, Nathan, Queensland, Australia
[2]School of Civil, Environmental and Mining Engineering, University of Adelaide, Adelaide, Australia
[3]Faculty of Environment and Natural Resources, Nong Lam University, Ho Chi Minh City, Vietnam
[4]School for Environment and Sustainability, University of Michigan, Ann Arbor, Michigan, USA
[5]Fenner School of Environment & Society, The Australian National University, Canberra, Australia
[6]Centre for Freshwater Ecosystems, La Trobe University, Wodonga, Victoria, Australia
[7]Department of Civil and Environmental Engineering, Princeton University, New Jersey, USA

*Correspondence to*: Songyan Yu (sunny.yu@griffith.edu.au)

## Abstract

There is a growing interest globally in the spatial distribution of intermittently flowing streams and rivers, and how their spatial extent varies in relation to climatic factors. However, deriving consistent information on the extent of flow intermittency within river networks is hampered by the fact that streamflow gauges are often sparsely distributed and more often be located within the most perennial parts of the river network. Here, we developed an approach to quantify catchment-wide streamflow intermittency over long timeframes and in a spatially explicit manner, using readily accessible and spatially contiguous daily runoff data from a national-scale water balance model. We examined the ability of the water balance model to simulate streamflow in two hydro-climatically distinctive (subtropical and temperate) regions in Australia, with a particular focus on low flow simulations. We also evaluated the effect of model time step (daily vs. monthly) on flow intermittency estimation to inform future model selection. The water balance model showed better performance in the temperate region characterised by steady baseflow than in the subtropical region with flashy hydrographs and frequent cease-to-flow periods. The model tended to overestimate low flow magnitude mainly due to overestimation of gains (e.g. groundwater release to baseflow) during low-flow periods. Modelled patterns of flow intermittency revealed highly dynamic behaviour in space and time, with cease-to-flow events affecting between 29 % and 80% of the river network over the period of 1911-2016, using a daily streamflow model. The daily flow model did not perform better than the monthly flow model in quantifying flow intermittency at a monthly time step, and model selection should depend on the intended application of the model outputs. Our general approach to quantifying spatio-temporal patterns of flow intermittency is transferable to other parts of the world, and can inform hydro-ecological understanding and management of intermittent streams where limited gauging data are available.

**Keywords:** AWRA-L model, flow regime, river routing, Australia, temporary streams

## 1 Introduction

Intermittent streams that cease to flow for some period of most years are prevalent within river networks globally (Acuña et al., 2014; Datry et al., 2014). Their spatial extent is projected to increase in regions experiencing drying trends related to climate change and water extraction for human uses (Larned et al., 2010). Intermittent streams have seen increasing research interest over the past decade (e.g. Costigan et al., 2016; Fritz et al., 2013; Gallart et al., 2017; Leigh et al., 2016), and there is a growing interest in conserving these unique ecosystems. The scarcity of spatially-explicit information on flow intermittency has been identified as one of the key issues confronting intermittent stream management (Acuña et al., 2017). Flow intermittency exerts primary control on the transfer of energy, materials and organisms by surface water through river networks (Jaeger et al., 2019) and is a key driver of riverine ecosystems (Datry et al., 2017; Poff et al., 1997; Stanley et al., 1997). Therefore, improved understanding of temporal and spatial patterns in flow intermittency is fundamentally important for effective river management.

Previous studies have predominantly relied on the use of gauged streamflow data to make inferences about the distribution of intermittent streams in many regions, including France (Snelder et al., 2013), Australia (Bond and Kennard, 2017; Kennard et al., 2010b), Spain and North America (de Vries et al., 2015). However, spatial biases in the distribution of stream gauges used in such studies may give misleading impressions of spatial patterns and extent of streamflow intermittency (Snelder et al., 2013). Alternative methods for quantifying the extent of intermittent flow include citizen-observation networks supported by regular reports from trained volunteers (Datry et al., 2016; Turner and Richter, 2011), the use of electrical arrays by measuring the electrical conductivity of the streambed (Jaeger and Olden, 2012), development of predictive models for intermittent streams (González-Ferreras and Barquín, 2017), and deployment of unmanned aerial systems (Spence and Mengistu, 2016). These alternatives are generally appropriate over small spatial extents and short time frames but are difficult to scale up to larger areas to quantify flow intermittency in space and time. Satellite remote sensing-based quantification of flow intermittency (Hou et al., 2019) can cover larger spatial extents, but for now, remains applicable only to relatively large rivers (> 30 m in the case of Landsat imagery) and can be affected by factors such as vegetation and cloud obstruction.

Spatially contiguous runoff data derived from water balance models provide another potential alternative to quantify spatio-temporal variations in flow intermittency. For example, Yu et al. (2018) used runoff simulations obtained from a water balance model WaterDyn (Raupach et al., 2009) to generate spatially explicit and catchment-wide estimates of streamflow intermittency, but only at a relatively coarse monthly time step. Depending on the application, flow simulations at a finer temporal scale (e.g. daily) may be necessary to capture the dynamic aspects of hydrological processes. These kinds of simulations are important to better understand the causes of flow intermittency at multiple spatial scales and enable ecologically-relevant characterisation of streamflow properties such as the magnitude, frequency, duration, and rate of change in high or low flow events. However, there are few examples of studies quantifying spatial and temporal variation in flow intermittency across river networks using spatially contiguous daily flow data. That is partly because streamflow simulation is

more challenging at a daily versus monthly time step due to higher uncertainties in input data at this finer temporal scale (Wang et al., 2011).

Water balance models at a daily time step have been increasingly developed around the world (Bierkens et al., 2015; Lin et al., 2019). One prominent regional example is the Australian Water Resource Assessment Landscape (AWRA-L) model (van Dijk, 2010). The AWRA-L model has been developed by the Commonwealth Scientific and Industrial Research Organisation

(CSIRO) and the Australian Bureau of Meteorology (BoM) to simulate the terrestrial water balance across Australia at a daily time step (Frost et al., 2016; van Dijk, 2010). The model yields spatially contiguous daily water availability values gridded at a spatial resolution of 0.05 arc-degree spatial resolution (approximately $5 \times 5$ km) (Frost et al., 2016). The development of such water balance models in Australia and other parts of the world provides the potential to quantify spatial and temporal variation in runoff, and hence flow intermittency, at a daily time step. However, this requires an effective and efficient

conversion process to translate gridded runoff estimates to accumulated streamflow estimates down the river network. This is especially challenging for large study areas due to lags in runoff, which can influence the timing of flow peaks and rates of recession. Additionally, many national-scale water balance models, including AWRA-L, were calibrated on a large domain that covers multiple climate conditions (Viney et al., 2015), providing a best "average" response but potentially inconsistent accuracy of runoff simulations within particular climate domains. As the predictive performance for ungauged basins strongly

depends on climate settings, this compromise raises the question as to whether such models can be used to quantify flow intermittency over multiple climate conditions. Although substantial efforts have been made in evaluating hydrological models in different climate conditions (Do et al., 2019; Gudmundsson et al., 2012; Lin et al., 2019; Zaherpour et al., 2018), a limited number of such studies have focused particularly on model performance during low flow conditions, which is particularly important for flow intermittency quantification.

In this study, we sought to apply spatially contiguous daily runoff outputs from the AWRA-L water balance model to quantify the spatial extent and temporal patterns of flow intermittency. To assess the accuracy of the AWRA-L model for daily flow simulations, we first developed a simple but effective technique to convert runoff to streamflow for two hydro-climatically distinctive regions. The translation of gridded runoff to aggregated streamflow/discharge on vector river flow lines make AWRA-L outputs more accessible to fluvial geomorphologists and ecologists, who may intend to relate daily hydrologic

characteristics of rivers to a broad range of physical and ecological phenomena. We further assessed the uncertainty of the AWRA-L model in capturing patterns of flow intermittency. Lastly, we evaluated the effect of time step (daily vs. monthly) on the relative performance of the model in replicating observed patterns of cease-to-flow periods at reference gauges. A previous study conducted at the monthly time step (Yu et al., 2018) was used to benchmark flow intermittency estimated from the AWRA-L model.

## 2 Study areas

This research was conducted in two hydro-climatically distinctive regions: South-east Queensland and the Tamar River catchment in Tasmania (Fig. 1). The South-east Queensland (SEQ) region is located in the eastern part of Australia (Fig. 1a) and comprises five major coastal river basins with a total area of 21,331 $km^2$ (Fig. 1b) (Australian Bureau of Statistics, 2011). SEQ has 7,229 stream segments and their corresponding sub-catchments according to the Australian Hydrologic Geospatial Fabric (Geofabric), with the minimum upstream drainage area of 1.5 $km^2$. SEQ is a region of transitional temperate to subtropical climate (Fig. 1a) with substantial inter- and intra- annual variation in rainfall. The majority of rainfall and streamflow usually occur in the summer months of January to March, often followed by a second minor discharge peak between April and June, but high and low flows may occur at any time of year (Kennard et al., 2007). Thus, there are a range of flow regimes with many streams being intermittent to varying degrees. The Tamar River catchment (Tamar) is located in Tasmania, an island state off Australia's south coast (Fig. 1a, c). It drains a catchment area of approximately 11,215 $km^2$, comprising over one-fifth of Tasmania's land mass and is located in north-east and central Tasmania. According to climate data from BoM (http://www.bom.gov.au/climate/data), Tamar is characterised by a temperate climate condition, of which rainfall is relatively evenly distributed throughout the year and most months receive very similar averages.

[Figure 1 is about here]

## 3 Data and Methodology

### 3.1 Streamflow gauge data

Gauged daily streamflow data were sourced from the BoM water data website (http://www.bom.gov.au/waterdata) and were used to assess accuracy of AWRA-L modelled streamflow (Section 3.3) and to estimate an appropriate zero flow threshold of modelled streamflow data for quantifying patterns of streamflow intermittency (Section 3.4). A total of 25 gauges in SEQ and 15 gauges in Tamar were selected (Fig. 1b, c) to assess modelled streamflow accuracy. These gauges had less than 0.5 % missing values over the period from 01/01/2005 to 31/12/2017 and had minimal hydrologic modification due to human activities. A larger set of 43 gauges in SEQ (including 21 of the 25 gauges used by us for streamflow validation) was used to estimate the zero-flow threshold for this region (see Yu et al., 2018 for details of stream gauges). The gauges were widely dispersed throughout each study area and encompassed a range of stream sizes, catchment areas (22 – 3,881 $km^2$ in SEQ; 33 – 3,294 $km^2$ in Tamar) and flow regime types, ranging from highly intermittent to perennial streams (see results). However, the set of stream gauges used in our analyses under-represented the frequency of small low-order streams in both regions. Therefore, we regard the selected gauges to be representative of the range of environmental and hydrological conditions in the regions, except for extremely small catchments with an area < 22 $km^2$ that likely have higher cease-to-flow occurrence.

## 3.2 Conversion from spatially contiguous runoff to streamflow

AWRA-L is a daily 0.05° grid-based distributed water balance model that is conceptualised as a small catchment. It simulates the water flow through the landscape from the rainfall entering the grid cell through the vegetation and soil and then out of the grid cell through evapotranspiration, surface water flow or lateral flow of groundwater to the neighbouring grid cells (Viney et al., 2015). AWRA-L was calibrated and validated at the national scale during its development by CSIRO and BoM, with 301 gauges used for calibration and a different set of 304 gauges used for validation (Zhang et al., 2013). Simulated daily

runoff from the AWRA-L model (version 5) was downloaded from BoM (http://www.bom.gov.au/water/landscape). These data are in gridded format and require conversion to streamflow for each sub-catchment by aggregating the gridded runoff data with a hierarchically nested catchment to simulate streamflow throughout river networks. The conversion process may or may not need to use a river routing model to propagate streamflow through river networks, partly depending on the size of the catchment of interest (Robinson et al., 1995). If streamflow simulated with a routing model shows little difference to that

without a routing model, then the conversion process is more efficient without a routing model, and the readily available runoff data can be more accessible for potential applications, such as flow characterisation for ungauged stream segments. In addition, a conversion process involving a routing model can be computationally-intensive and usually requires parallel computing to speed up the calculations (David et al., 2011b). Therefore, in this study, we applied two approaches to determine an effective and efficient runoff-streamflow conversion. The first approach coupled a river routing model to the water balance model, and

its effects on flow simulations are compared to the model performance of a lumped model, which was operated without any river routing (Fig. 2). As the conversion process was achieved using the "*catchstats*" package (https://github.com/nickbond/catchstats) in the R programming language (R Development Core Team, 2017), so the second approach was to speed up the conversion process by incorporating parallel algorithm to exiting functions of that package. The conversion process was run on a Griffith University High-Performance Computing node with 12 cores and RAM 12 GB.

[Figure 2 is about here]

The hierarchically nested catchment dataset used in this study was sourced from the Geofabric dataset (Stein et al., 2014), which provides a fully connected and directed stream network derived from the national 9 arc-second DEM and flow direction grid (~250m resolution), and associated catchment hierarchy at the national scale. The routing model applied in this study was the Routing Application for Parallel computatIon of Discharge (RAPID) model (David et al., 2011b). RAPID solves the matrix-

based Muskingum equation to route flow through each stream of the river network and performs streamflow computation for every stream segment of a river network, including ungauged streams. Various water balance models have been used in combination with RAPID (Follum et al., 2017; Lawrence et al., 2011; Lin et al., 2019).

To test the effects of river routing, we first calculated a series of flow metrics (Table 1) for flow simulations from both the lumped and coupled models. The calculated flow metrics are commonly used to describe the critical components of flow

regimes across average, high, and low flow conditions, including flow magnitude and variability, the timing, frequency and duration of high and low flows, and rates of changes in flow events (Olden and Poff, 2003; Poff et al., 1997). Calculation of these streamflow characteristics allows a comprehensive assessment of the effects of river routing on streamflow simulations in the two regions. We then applied Wilcoxon rank sum test for each flow metric to determine whether the inclusion of river routing can improve model accuracy based on a significance level of 5 %. We used the 10th and 90th percentiles of daily flows to respectively describe low-flow and high-flow thresholds (Gudmundsson et al., 2019; Leigh and Datry, 2016). The calculation process was conducted with the "*hydrostats*" package in the R language (Bond, 2016).

[Table 1 is about here]

### 3.3 Accuracy assessment of modelled streamflow

To evaluate overall model performance in streamflow simulations, we calculated the modified Kling-Gupta efficiency (KGE; Kling et al, 2012) between the observed and modelled streamflow for all gauges in SEQ and Tamar (Eq. (1)). KGE is an integrated skill metric, which measures the Euclidean distance between a point and the optimal point that has the maximum correlation coefficient, zero variability error and zero bias error between the simulated and observed streamflow (Gupta et al., 2009; Kling et al., 2012). KGE takes values from -1 to 1: KGE = 1 indicates perfect agreement between simulations and observations, and KGE < -0.41 indicates that the mean of observations provides better estimates than simulations (Knoben et al., 2019). To evaluate model performance in different components of flow regimes, we also calculated each summary flow metric (Table 1) for observed and modelled streamflow data at all gauges in SEQ and Tamar and visually compared their frequency distributions. The use of KGE provides an overall assessment of AWRA-L model performance and the flow metrics in Table 1 are used to comprehensively evaluate the model accuracy for various components of flow regimes, including the flow metrics related to low flows. Only six of the 25 gauges in SEQ and three of the 15 gauges in Tamar were the same as those used to calibrate the AWRA-L water balance model. This small overlap between the AWRA-L calibration gauge set (n=301) and the streamflow model validation gauge set (n=25 in SEQ and 15 in Tamar) means that potential overestimation of streamflow model performance is likely to be minimal.

$$KGE = 1 - \sqrt{(r-1)^2 + (\beta-1)^2 + (\gamma-1)^2}$$ (Equation 1)

$$\beta = \frac{\mu_s}{\mu_o}; \; \gamma = \frac{CV_s}{CV_o} = \frac{\sigma_s/\mu_s}{\sigma_o/\mu_o}$$

where $KGE$ is the modified KGE-statistic (dimensionless), $r$ is the correlation coefficient between simulated and observed runoff (dimensionless), $\beta$ is the bias ratio (dimensionless), $\gamma$ is the variability ratio (dimensionless), $\mu$ is the mean runoff in

m$^3$s$^{-1}$, $CV$ is the coefficient of variation (dimensionless), $\sigma$ is the standard deviation of runoff in m$^3$ s$^{-1}$, with subscripts $s$ and $o$ referring to simulated and observed runoff values, respectively.

Furthermore, considering that this study aims to apply flow simulations to quantify flow intermittency, the model accuracy of low flow simulation is particularly important. The study period (01/01/2005-31/12/2017) was considered sufficient to assess low flows. The 13 year study period is close to a discharge record length of 15 years which Kennard et al. (2010a) concluded is sufficient to enable accurate estimation of low flow metrics. In addition, our study period begins in the middle of the Australian Millennium Drought (2001-2009), and includes a significant low-flow period. A preliminary analysis showed that AWRA-L modelled streamflow was sensitive to rainfall events, relative to the response of observed flow (Fig. 3). This finding indicates that over-responsiveness of AWRA-L to rainfall may potentially contribute to overestimation of low flow. We hypothesised that this over-responsiveness is partly due to overestimation of "*in situ*" gains to low flow discharge (e.g. groundwater release to baseflow) as well as underestimation of transmission losses (e.g. depression filling and evapotranspiration) during water movement through various flow paths in the stream network (Davison and van der Kamp, 2008). Given that we do not have access to the AWRA-L model to directly adjust model parameters, we instead compared the observed and modelled low flow magnitude at all gauges in the two study areas along the gradient of their catchment areas (22-3,881 km$^2$ in SEQ; 33-3,294 km$^2$ in Tamar) to test this hypothesis. We expect that 1) if the difference in low flow magnitude occurs at all gauges, then low flow overestimation can be at least attributed to the overestimation of gains to low flow discharge. Alternatively, 2) if the difference in low flow magnitude occurs towards the downstream of the catchment, then low flow overestimation may be related to underestimation of transmission losses.

[Figure 3 is about here]

### 3.4 Quantifying flow intermittency using spatially contiguous flow simulations

Given the fact that water balance models often over-predict the magnitude of very low flows due to the difficulties of quantifying hydrological processes influencing low flow discharge (Smakhtin, 2001; Staudinger et al., 2011; Ye et al., 1997), we adopted the same method used in Yu et al. (2018) to estimate a threshold of zero flow from the model that related measured zero-flow duration at each gauge to catchment environment variables. We used linear regression to model the mean annual zero flow duration (daily time step) at each gauge as a function of catchment environment variables. This regression analysis was only conducted in SEQ as most gauges in the Tamar catchment had perennial flow. The environmental variables were the same as those in Yu et al. (2018), and included variables related to climate (annual daily maximum temperature), catchment geology topography (catchment area, catchment average slope, and catchment average elevation), and catchment soil properties (catchment average saturated hydraulic conductivity). Regression models were developed using all possible predictor variable combinations and we selected the "best" model for predicting zero flow duration based on corrected Akaike's Information Criterion (AICc) (Hurvich and Tsai, 1989). To estimate the prediction error of the selected model, we applied

leave-one-out cross validation on the selected 43 gauges and reported prediction error ($R^2$) to estimate the model prediction performance. Regression model development and cross-validation were conducted with the *MuMIn* and *boot* packages in R

(R Development, 2017). Regression analyses were performed on all combinations of predictor variables and the best model with the lowest AICc (-54.2) retained five covariates, including annual daily maximum temperature, catchment area, slope, average elevation, and average saturated hydraulic conductivity. The developed predictive model showed a good model fit with an adjusted $R^2$ of 0.71, and the leave-one-out cross validation on the regression model showed relatively good model performance with an average $R^2$ of 0.64. We checked for spatial autocorrelation of the regression model residuals (as

recommended by Dormann et al., 2007) and found they were not significantly autocorrelated (Moran's I = -0.06, p = 0.69). Examination of spatial residual maps further supported this conclusion, with no spatial trends in model residuals apparent.

Next, we used the predictive models to extrapolate estimates of overall flow intermittency (in terms of the proportion of days with zero flow) to each segment throughout the river network. Finally, for each segment, the time-series of daily runoff was truncated (flows below the threshold were set to "0") by adopting an appropriate threshold of "zero flow" that preserved the

proportion of days with flow as estimated in the previous step. The adopted thresholds ranged from $0 – 1.668$ m$^3$/s, with a median value of 0.002 m$^3$/s. We recognise several sources of uncertainty in our approach to estimating the zero flow thresholds. The unexplained variation in the predictive model may be due to the limited number of environmental attribute covariates used in the model and hence ability to adequately represent the range of environmental processes that influence streamflow intermittency. Additional uncertainty in model predictions may arise because the distribution of stream gauges

used for model calibration under-represented the frequency of extremely small catchments that likely had higher cease-to-flow occurrence.

Based on the modelled daily streamflow from AWRA-L, we calculated annual flow intermittency as the number of zero-flow days per year over the period of 2005-2016. To evaluate the effect of time step (daily vs. monthly) on the relative performance of AWRA-L in replicating observed patterns of cease-to-flow periods, we compared model outputs with those derived from a

monthly water balance model – the WaterDyn model (Fig. 2). Monthly flow intermittency estimated from WaterDyn was thus used to benchmark results from the monthly AWRA-L. To do this, we aggregated daily outputs to a monthly time step (termed "monthly AWRA-L" hereafter, Fig. 2). We tried two different aggregation methods. One considered that the flows for a month were zero when at least one day in that month had zero flow (termed "monthly AWRA-L_01" hereafter), and the other considered that all days in a month must have zero flow for that month to be zero (termed "monthly AWRA-L_30" hereafter).

These two methods together should provide both upper and lower bounds of comparing daily and monthly models in estimating flow intermittency. The WaterDyn model was developed to provide monthly spatially contiguous water balance data at the Australian continental scale by CSIRO and BoM with a similar model structure to AWRA-L (Raupach et al., 2018), and has been used to quantify the spatial and temporal patterns of flow intermittency in SEQ following similar methods to this study (Yu et al., 2018). Modelled flow intermittency from all three sources (i.e. daily and monthly AWRA-L, and monthly

WaterDyn) was also tested against the measured flow intermittency derived respectively from daily and monthly observed streamflow data at gauged locations in SEQ.

Taking advantage of the modelled long-term runoff data from AWRA-L over the period of 1911-2016, we further quantified spatial and temporal dynamics of flow intermittency for every stream segment within SEQ, and compared the results with those from the WaterDyn model over the same period (Yu et al., 2018). The spatial pattern of flow intermittency was

represented by the mean annual number of zero flow days across the period of 1911-2016 for AWRA-L and by the mean annual number of calendar months for WaterDyn. The temporal pattern of flow intermittency was expressed as the proportion of streams with flow intermittency > 30 days or 1 month (termed "intermittent streams" hereafter) for AWRA-L and WaterDyn, respectively.

## 4 Results

### 4.1 Negligible effects of river routing on daily flow simulations

The lumped and coupled (i.e. with routing) models using AWRA-L simulated runoff were run in both SEQ and Tamar, and produced similar values for various flow metrics between the lumped and coupled in both regions (Fig. 4; $p$ values were greater than 0.50 for most flow metrics based on Wilcoxon test results). There were noticeable but not statistically significant differences for two flow metrics related to low flows (the variability in timing, and the frequency of low flow spells), and only

the duration of low flow spells was statistically significant $(p = 0.03)$. These results suggested that the routing algorithm has nearly negligible effects on flow simulations in our study areas, which is reasonable because of the small size of the two watersheds. Therefore, in the subsequent analysis, we only used the results from the AWRA-L lumped model as it is relatively less computationally intensive and was able to maintain a comparable model performance to that of the coupled model taking into account the routing effect.

[Figure 4 is about here]

### 4.2 Accuracy assessment of modelled streamflow in SEQ and Tamar

The overall accuracy of streamflow estimated by AWRA-L lumped model (referred to as "modelled streamflow" in this section) was evaluated for 25 gauges in SEQ and 15 gauges in Tamar. Results suggested a fair to good explanatory value across all gauges (Fig. 5). The KGE values varied across the 25 gauges in SEQ, ranging from -0.19 (Gauge No. 145103) to

0.76 (143901), with a median value of 0.42, while the model generally performed better in Tamar and the KGE values ranged from 0.11 (18219.1) to 0.71 (852.1) across 15 gauges, with a median value of 0.47 (Fig. 5). However, no significant difference was found in the overall model performance between the two hydro-climatically distinctive regions, according to Wilcoxon test $(w = 247, p = 0.10)$.

[Figure 5 is about here]

Concerning model performance in simulating different components of flow regimes, the modelled streamflow in SEQ revealed a generally good match with the observed streamflow across all high-flow metrics and the magnitude of average flow, but the model tended to overestimate the variation in the magnitude of average flow (almost two times higher on average), report earlier timing of low flows, overestimate the frequency (48 % higher), and underestimate the duration (74 % lower) of low flows (Fig. 6). Compared to the model performance in SEQ, the flow simulations in Tamar showed slightly better performance,

predicting well not only for the high-flow metrics but also for the metrics related to average flows (Fig. 6). However, flow simulations in Tamar also exhibited slightly earlier estimations for the timing of low flow spells (13 % earlier), overestimations for low flow spell frequency (92 % lower on average) and underestimation for low flow spell duration (58 % lower) (Fig. 6).

[Figure 6 is about here]

Varying degrees of difference in the magnitude of low flow between the observed and modelled were found among the gauges.

There appeared to be a tendency toward larger differences with increasing catchment area in SEQ but not in Tamar (Fig. 7). The models appeared to over-estimate "*in situ*" gains to low flow in some reaches in both regions, while under-estimating transmission losses in SEQ, suggesting that over-estimation of "*in situ*" gains in AWRA-L likely contribute to the overall overestimation of low flow in downstream catchments.

[Figure 7 is about here]

**4.3 Quantifying flow intermittency using flow simulations**

We calculated annual flow intermittency at gauged locations in SEQ using three sources of modelled flow (daily and monthly AWRA-L, and monthly WaterDyn). Annual flow intermittency calculated using daily AWRA-L flow (i.e., the average number of cease-to-flow days per year) was tested against annual flow intermittency estimated using observed data (Fig. 8a). The AWRA-L model displayed the potential to be used to estimate flow intermittency at a daily time step, with a fair match with

the observed flow intermittency ($R^2$= 0.56) in SEQ. Nonetheless, the model tended to overestimate flow intermittency for gauges located in relatively wet areas (e.g. ≤ 40 days of flow intermittency per year) while underestimating for gauges located in relatively dry areas (e.g. ≥ 40 days of flow intermittency per year).

Figure 8b shows annual flow intermittency calculated using monthly AWRA-L flow and monthly WaterDyn flow. In this case, annual flow intermittency was defined as the average number of months characterized with zero flow. The WaterDyn model

showed much more accuracy than the two aggregation methods based on the monthly AWRA-L model in estimating flow intermittency ($R^2$ = 0.53, 0.43 and 0.32 respectively for monthly WaterDyn, monthly AWRA-L_01 and monthly AWRA-L_30). More specifically, the WaterDyn model displayed a similar estimation pattern as the daily AWRA-L model:

overestimation in relatively wet areas while underestimation in relatively dry areas. By contrast, not surprisingly, the two aggregation methods showed the upper and lower bounds of flow intermittency estimates from the monthly AWRA-L model: monthly AWRA-L_01 overestimated flow intermittency and monthly AWRA-L_30 underestimated flow intermittency at nearly all gauges (Fig. 8b).

[Figure 8 is about here]

The spatial patterns of flow intermittency derived from the daily AWRA-L and monthly WaterDyn flow simulations aligned well for the main stems and some coastal streams, which were predicted to flow for most of the time (Fig. 9a, b). There were noticeable spatial differences between model predictions of streamflow intermittency for low order inland streams. For example, in the western Brisbane River catchment and the South Coast River catchment, most inland streams were predicted by the daily model to flow for longer period than by the monthly model; while in the Pine River catchment and the Logan-Albert River catchment, many inland streams were predicted by the daily model to flow for a shorter period (Fig. 9a). Compared to the monthly WaterDyn model, fewer streams were predicted by the daily AWRA-L to experience extremely long dry events as well as less than one month of zero flows (Fig. 9c, d). However, more streams on average (60 % vs. 49 % for the AWRA-L and WaterDyn model, respectively) were predicted to flow intermittently (> 30 days or > 1 month) to varying degrees in SEQ, which suggests that flow intermittency was prevalent in SEQ, irrespective of the water balance model used.

[Figure 9 is about here]

Temporally, the daily model estimated that the proportion of intermittent streams in SEQ varied from 29 % to 80 % over the study period (1911-2016), while the monthly model estimated the range to be from 3 % to 80 % estimated during the same time span (Fig. 10). The two temporal patterns were temporally correlated (r = 0.71) and similar predictions with higher proportions of intermittent streams were estimated for the dry years by both models. Compared to dry years, the two models exhibited greater differences in predictions for the wet years, where the daily model tended to predict more proportion of intermittent streams. Overall, the daily model suggested a drier history in SEQ in terms of flow intermittency than the monthly model. The models successfully identified the extensive drying associated with severe drought periods. Notably, the Widespread drought (1914-1920), WWII drought (1939-1946) and Millennium drought (2001-2009) were all visible in both two sets of model predictions.

[Figure 10 is about here]

## 5 Discussion

The scarcity of information on the spatial and temporal extent of flow intermittency has been identified as a major barrier for ecologists and managers to understand and protect intermittent stream ecosystems (Acuña et al., 2017). This barrier has been

partly overcome in previous studies by using statistical models relating flow intermittency to surrounding environmental variables (Bond and Kennard, 2017; González-Ferreras and Barquín, 2017; Jaeger et al., 2019; Snelder et al., 2013), but most of these studies focused on only the spatial variations in flow intermittency, except for Jaeger et al. (2019), overlooking its temporal aspects. This issue becomes particularly urgent in the time when flow regimes of streams are changing worldwide, mainly in response to climate change and water extraction for human uses (Chiu et al., 2017; Jaeger et al., 2014). Monthly runoff data have been recently used to quantify flow intermittency for entire river networks (Yu et al., 2018), and the current study takes one step further to use daily runoff data in flow intermittency estimation, which is especially needed for studies aimed at quantifying ecological responses to short term flow events (e.g. frequent zero flow events within a month). In this study, we comprehensively examined the ability of a daily water balance model to simulate streamflow, with a particular focus on low flow simulations. We also investigated how to better choose water balance models to estimate flow intermittency by answering the question that whether daily flow models outperform monthly flow models at both daily and monthly scales. Our study can not only inform the estimation of the spatial distribution of intermittent flow but also reveal the temporal dynamics of intermittent streams over long timeframes.

**5.1 Efficient runoff-streamflow conversion for eco-hydrological research**

Effects of river routing on daily flow simulations were found negligible in SEQ and Tamar, most probably due to the relatively small size of the two catchments and the relatively short length of even the longest streams (Cunha et al., 2012). This can be verified with the concept of time of concentration, which is commonly used to measure the time needed for water to flow from the most remote point in a catchment to the catchment outlet. By following the formula for calculating the time of concentration proposed by Pilgrim and McDermott (1982) that has been widely used in Australia, we found the time of concentration in SEQ is around 33 hours, only slightly more than a daily time step (24 hours). This illustrates why it is difficult for a daily time-step routing model to effectively capture routing lags in our study domain. Negligible effect of river routing on flow simulations was also observed in previous studies (David et al., 2011a). Robinson et al. (1995) found that catchment size is a primary factor to determine which process, the hillslope or the channel network transport component, characterise lags in catchment runoff down the river network. In areas such as SEQ and Tamar that have a relatively small catchment size, the inclusion of channel network transport contributes little to the improvement of flow simulations. The negligible effect of river routing in SEQ and Tamar allowed us to simplify the simulation of daily flows without coupling with a river routing model. Hence we were able to use existing runoff outputs from the daily AWRA-L model. Arguably, similar opportunities exist in other small catchments.

**5.2 Accuracy assessment of modelled daily streamflow in two hydro-climatically distinctive regions**

Daily streamflow estimates showed a fair to good overall alignment with the observed flows in both SEQ and Tamar, with all gauges showing that flow simulations were better estimates than the mean of observations (KGE ≥ -0.41 at all gauges). Interestingly, although streamflow was more accurately simulated in the Tamar than in SEQ (the median values of KGE were

0.47 and 0.42, respectively), the differences between the two hydro-climatically distinctive regions were relatively small.

Despite ongoing efforts to calibrate AWRA-L against a set of reference scales distributed across the continent (Viney et al., 2015), this finding was reassuring given the much higher variability in rainfall and soil moisture in SEQ, factors that typically can lead to a more nonlinear streamflow response to rainfall (Poncelet et al., 2017), which possibly undermines the ability of water balance models to reliably predict runoff (Sheng et al., 2017). These results hence bode well for the application of AWRA-L outputs across diverse hydroclimatic regions.

When looking into the model performance for specific components of the flow regime, average- and high-flow metrics were both modelled well in Tamar, while only high-flow metrics were modelled well in SEQ. However, in both regions, the AWRA-L model showed poor performance in low flow metrics: overestimating the frequency and underestimating the duration of low flows, consistent with previous studies (Costelloe et al., 2005; Ivkovic et al., 2014; Staudinger et al., 2011; Ye et al., 1997). This suggests that the AWRA-L model is a generally robust model in predicting average- and high-flows, but still needs some

improvement to better simulate low flows. Runoff generation processes can vary substantially through space and time due to such factors as variations in soil depth, antecedent soil moisture and groundwater connectivity, and this can influence spatio-temporal variations in low flow characteristics, including streamflow intermittency (Zimmer and McGlynn, 2017). However, it is unknown the extent to which this contributed to uncertainty in the simulation of low flows and estimation in streamflow intermittency in this study. The uncertainty of AWRA-L in low flow simulations can be linked to its over-responsiveness to

rainfall, partly caused by overestimation of "*in situ*" gains and underestimation of transmission losses to low flow discharge, as shown in SEQ. Previous studies found that lateral flow exchange between grid cells of land surface models (e.g. AWRA-L) plays a significant role in redistributing soil water (Kim and Mohanty, 2016), and thus may improve "*in situ*" surface/subsurface runoff simulations (Lee and Choi, 2017). On the other hand, hydrological processes involved in transmission losses have been extensively discussed (Jarihani et al., 2015; Konrad, 2006), and studies have developed methods

to calculate transmission losses for better flow simulations (Costa et al., 2012; Lange, 2005). Therefore, low flow simulations by AWRA-L can possibly be improved by incorporating lateral flow exchange algorithms and better accounting for hydrological process such as evapotranspiration from riparian vegetation and infiltration into channel beds. This improvement is made more likely as AWRA-L has been released as a Community Modelling System (https://github.com/awracms/awra_cms), which allows co-development by the research community.

**5.3 Choose appropriate water balance model to quantify spatio-temporal dynamics of flow intermittency**

Our results suggest that the temporal resolution of analysis should be dictated by the resolution of input streamflow data. More specifically, the daily AWRA-L flow showed promise for estimating flow intermittency at a daily time step, while the monthly WaterDyn model was better than the monthly AWRA-L model in flow intermittency estimation at a monthly time step. This suggests that monthly flow models can sometimes outperform daily flow models in quantifying flow intermittency, depending

on the intended temporal resolution of the analysis. For example, daily flow models may be appropriate for studies aimed at

quantifying ecological responses to short term flow events, while monthly flow models are more suitable for research requiring the average degree of flow intermittency at a large spatial or temporal scale, such as examining the effect of flow intermittency on aquatic/streamside vegetation or species distributions (Stromberg et al., 2005). In addition, our study also suggested that the suitability of a monthly model (WaterDyn) for monthly resolution of analysis was not challenged by a daily model (AWRA-L) simply through aggregating daily streamflow simulations to a monthly time step. The aggregation methods used here applied one day or 30 days as a threshold and, respectively, either substantially overestimated or underestimated flow intermittency.

Spatially contiguous runoff data were used in this study to quantify spatial and temporal dynamics of flow intermittency, shedding light on the temporal aspect of flow intermittency that has been often overlooked in previous studies. Annual flow intermittency in SEQ was shown to vary significantly from year to year, ranging from 29 % to 80 % of total stream length for the AWRA-L model. However, given the limited spatial resolution of the Geofabric stream network data (9 arc-second longitude-latitude resolution, with the minimum upstream drainage area of 1.5 km$^2$) and hence ability to resolve the smallest streams, and that small streams are more likely to be intermittent, the proportion of predicted intermittent streams in SEQ may be under-estimated in our study. Although there are differences in the temporal patterns of estimated flow intermittency between the AWRA-L and WaterDyn models, neither model estimated intermittency to have a clear trend over the past century. However, there is still the concern about the potential shift of some perennial streams to intermittent streams due to climate change and intense human activities, as it has been evident in several regions where the number of low-flow and/or no-flow days is increasing (King et al., 2015; Ruhí et al., 2016; Sabo, 2014). Jaeger et al. (2014) investigated the effect of climate change on flow intermittency patterns and found that annual zero-flow days frequency were projected to increase by 27 % by mid-century in the Lower Colorado River Basin of United States. Research looking into projected changes in regional climate regimes can provide insights into future scenarios people may face, but such research is still scarce.

The approach developed here to generate spatially continuous estimates of streamflow characteristics (including flow intermittency) throughout stream networks has potential applicability to other regions of Australia and globally. All the data used in this study are available for the Australian national scale, and similar datasets also exist in other countries. For example, similar to the Geofabric data (Stein et al., 2014) used here, the National Hydrography Dataset Plus (NHDPlus) and HydroATLAS (Linke et al., 2019) provide hydrographic datasets and hydro-environmental attributes for national (USA) and global scales, respectively. In addition, similar to the daily flow model AWRA-L used in this study, other global and national-scale hydrologic models are also available, such as the global WaterGAP model (Döll et al., 2003), the community Noah land surface model (Noah-MP) (Niu et al., 2011) in the US and the HYPE model (Lindström et al., 2010) in Sweden.

## 6 Conclusions

In this study, we presented an approach to quantifying spatially explicit and catchment-wide flow intermittency over long timeframes based on spatially contiguous daily runoff data from a readily accessible water balance simulation. This research

builds upon previous studies using monthly runoff data, and paves the way for ecological research looking for metrics of flow intermittency at a daily time step. By testing this approach in eastern Australia, we not only confirmed our previous finding that intermittent flow conditions prevailed in the majority of streams, but also provided more detailed information on their

spatio-temporal variability at a daily time step. The proposed approach has potential applicability to other catchments globally, but our results also highlighted some complexities that future research should address to help improve the reliability of model outputs.

## Data availability

The data used in this study are available publicly online and the access websites have been listed in the main text where they

were first mentioned.

## Competing interests

The authors declare that they have no conflict of interests.

## Author contribution

AvD, HXD, MK and SY designed the research, and SY and HXD carried it out. SY wrote the original draft, and HXD, AvD,

PL, NB and MK contributed to writing of subsequent drafts.

## Acknowledgements

The project was supported by the Australian Climate and Water Summer Institute organised by the Australian Energy and Water Exchange Research Initiative (OzEWEX) and partners. This research was undertaken on the NCI National Facility in Canberra, Australia, which is supported by the Australian Commonwealth Government. We also gratefully acknowledge the

support of the Griffith University eResearch Services Team and the use of the High Performance Computing Cluster "Gowonda" to complete this research. SY was financially supported by the China Scholarship Council and Griffith University. HXD is currently funded by School for Environment and Sustainability, University of Michigan through grant number U064474. We would like to thank the Editor Christa Kelleher for handling our submitted manuscript, and thank George Allen and three anonymous reviewers for their comments and suggestions that much improved the final manuscript.

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

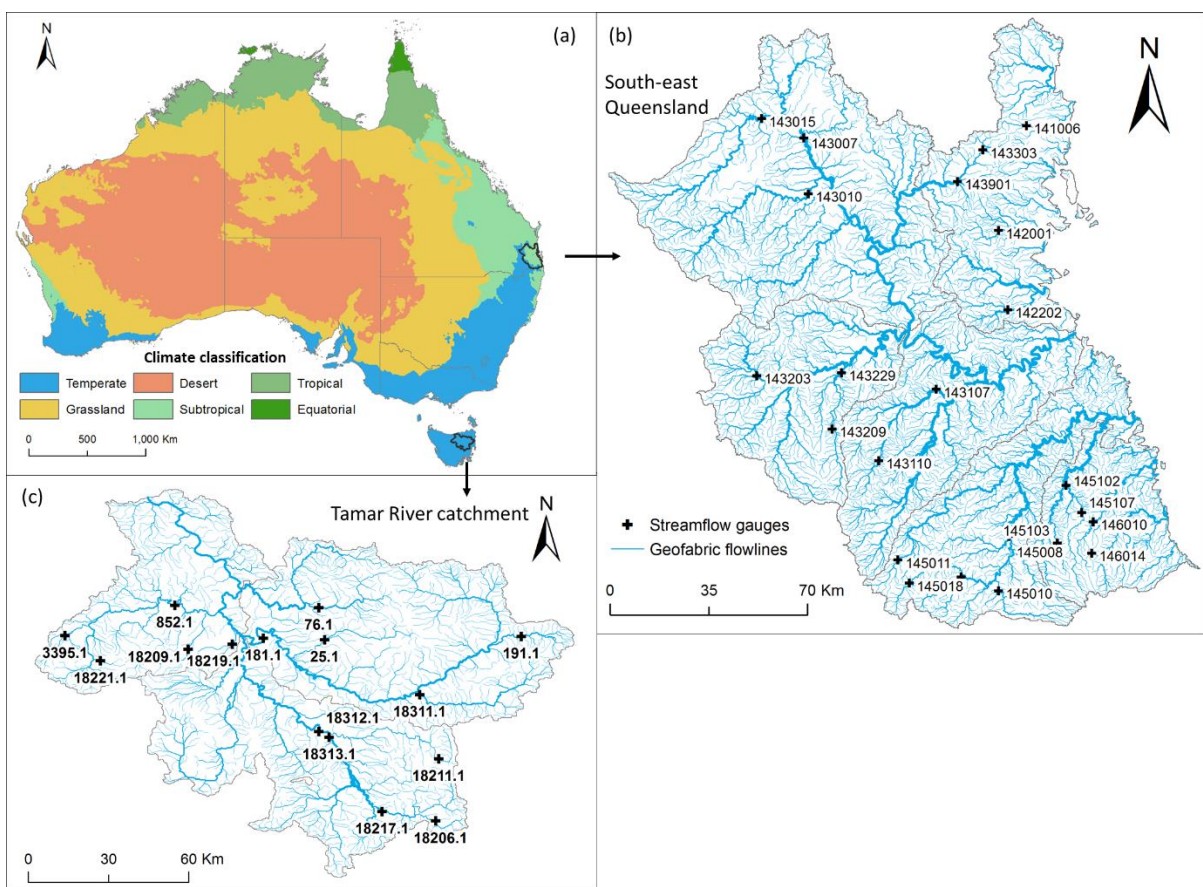

**Figure 1. Locations of the two climatically and hydrologically distinctive regions in Australia (a): South-east Queensland (SEQ) (b) and the Tamar River catchment (Tamar) (c) with Geofabric river networks and selected stream gauges (25 and 15 gauges for SEQ and Tamar, respectively). The climate classification in (a) is based on the Köppen classification system (Australian Bureau of Meteorology, 2014).**


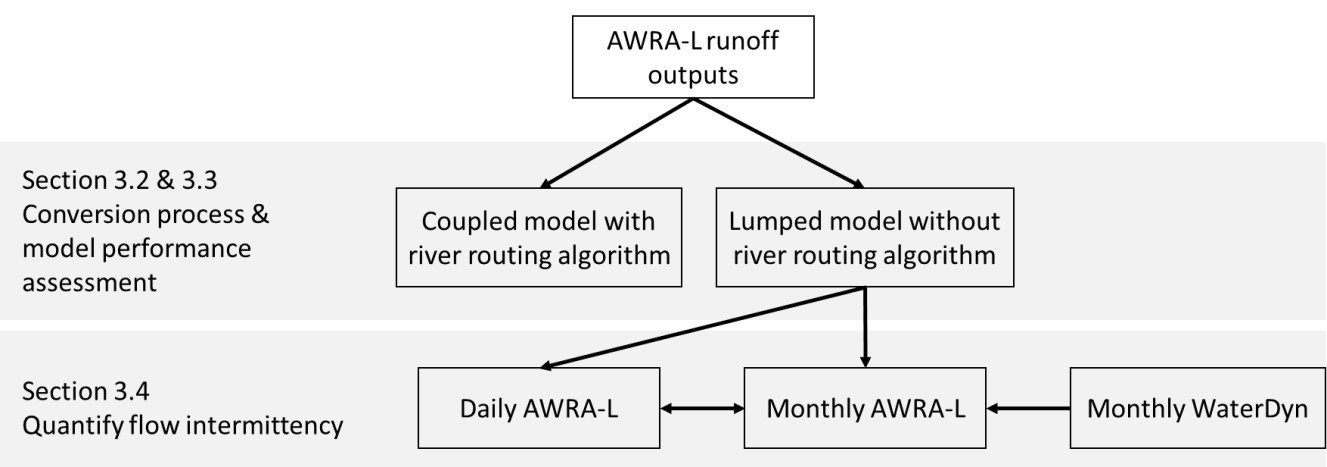

**Figure 2. Model configurations and their applications in this study. AWRA-L runoff outputs are translated to accumulated streamflow estimates with river routing algorithm (coupled model) and without (lumped model). These two model configurations are applied to test the effect of river routing on streamflow simulation accuracy. Based on the lumped model, we simulate daily streamflow throughout river networks (Daily AWRA-L) and further convert the daily stimulations to monthly outputs (Monthly AWRA-L). Both simulations are used to quantify streamflow intermittency, while results from a different monthly model (Monthly**
**WaterDyn) are used to benchmark the flow intermittency estimates from Monthly AWRA-L.**

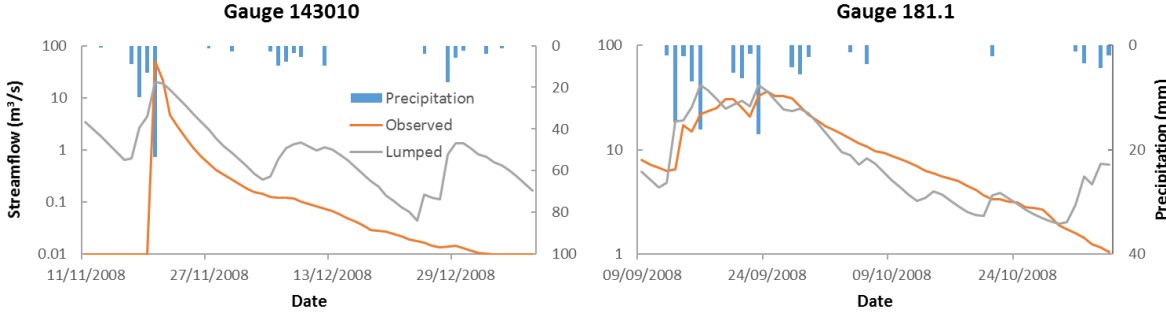

**Figure 3. Comparison of the observed and modelled hydrograph with the rainfall time series at gauges 143010 in SEQ and 181.1 in Tamar. The over-responsiveness of the model to rainfall is illustrated in the noticeable increase in modelled streamflow when a rainfall event occurred, while there is no obvious increase in observed streamflow. Rainfall data were sourced from the AWRA-L input.**


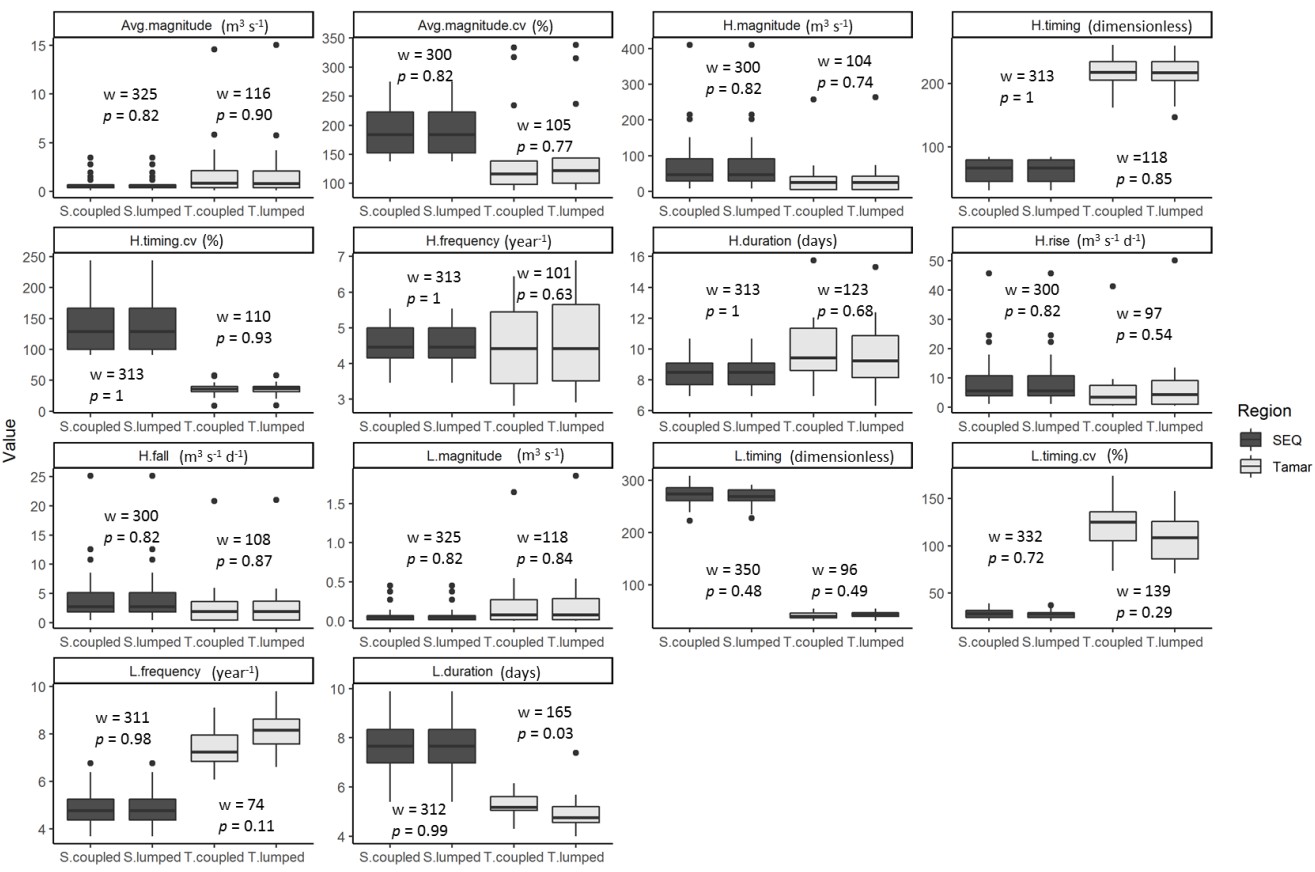

Figure 4. Comparison of hydrological characteristics between the lumped and coupled models in SEQ and Tamar. Refer to Table 1 for measurement description for each flow metric. Metrics are grouped according to average (Avg), high (H) and low (L) flow conditions. The values of *w* statistic and associated *p* values are also shown to indicate whether there is any significant difference between the coupled and lumped simulations.

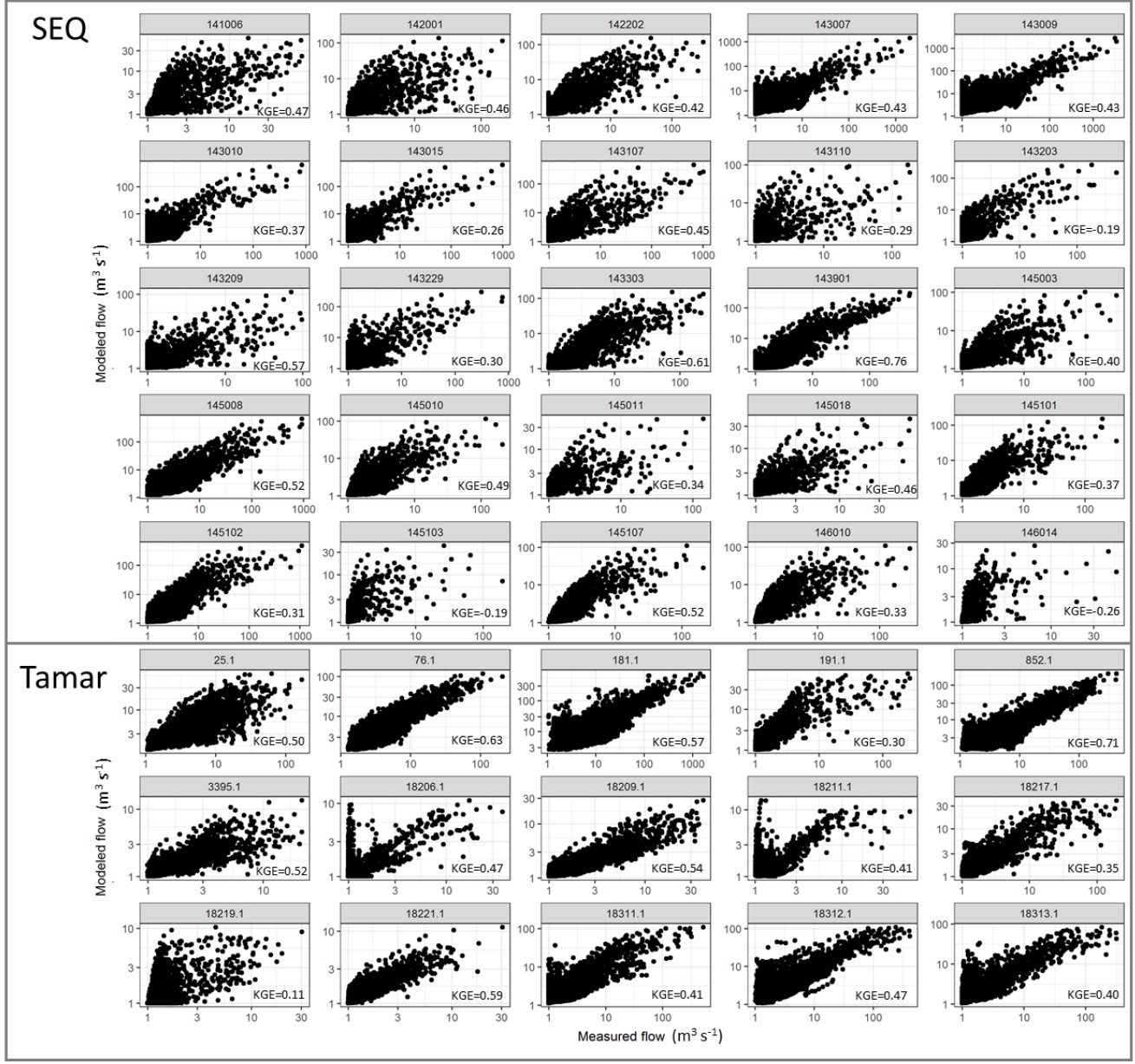

**Figure 5: Scatter plots of the measured and modelled (lumped) streamflow for each gauge station in SEQ and Tamar. The modified Kling-Gupta efficiency (KGE) is presented in each panel. The x and y axes are log-transformed ($\log_{10}(x+1)$) to aid interpretation.**

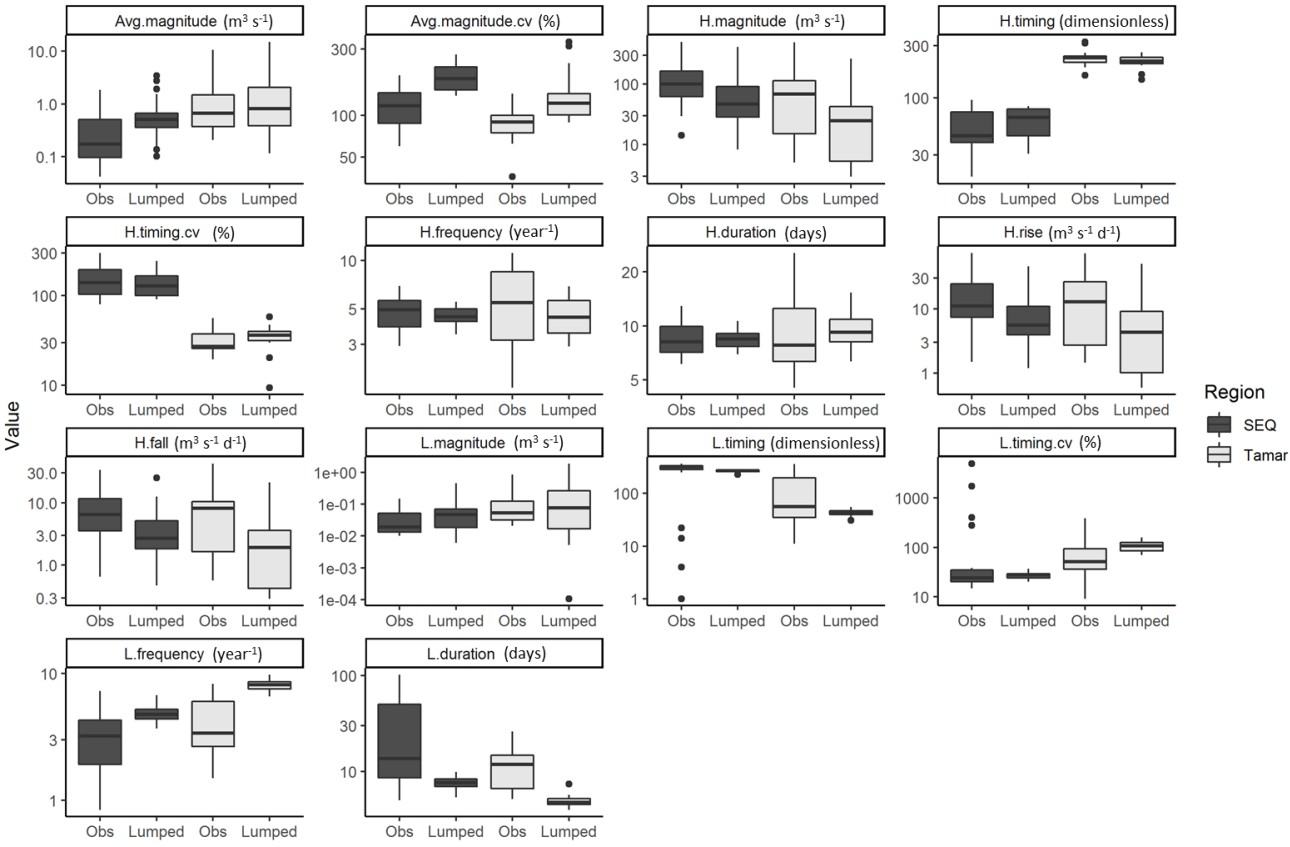

Figure 6. Variation in observed and modelled (lumped) hydrologic characteristics in SEQ and Tamar (n= 25 and 15 gauge locations, respectively). Refer to Table 1 for measurement description for each flow metric. Metrics are grouped according to average (Avg), high (H) and low (L) flow conditions. The y-axis is on a log scale for better interpretation.

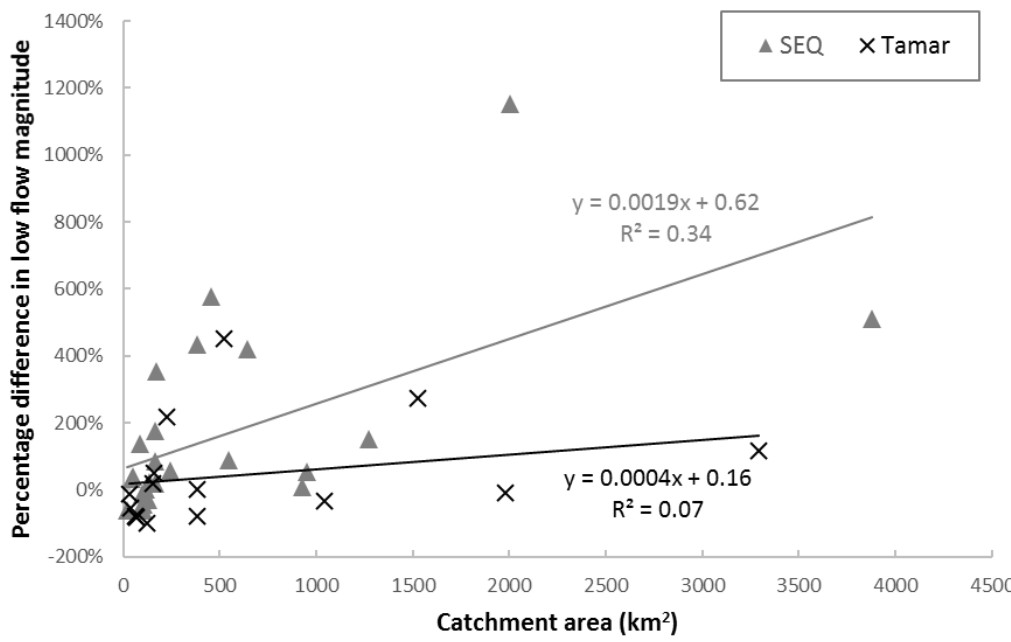


**Figure 7. Scatter plot of gauged catchment areas and percentage difference in low flow magnitude between the observation and simulation in SEQ (solid grey triangle) and Tamar (black cross). The regression line for each region is also shown as solid line (grey line for SEQ and black line for Tamar) with the regression function and $R^2$ value.**

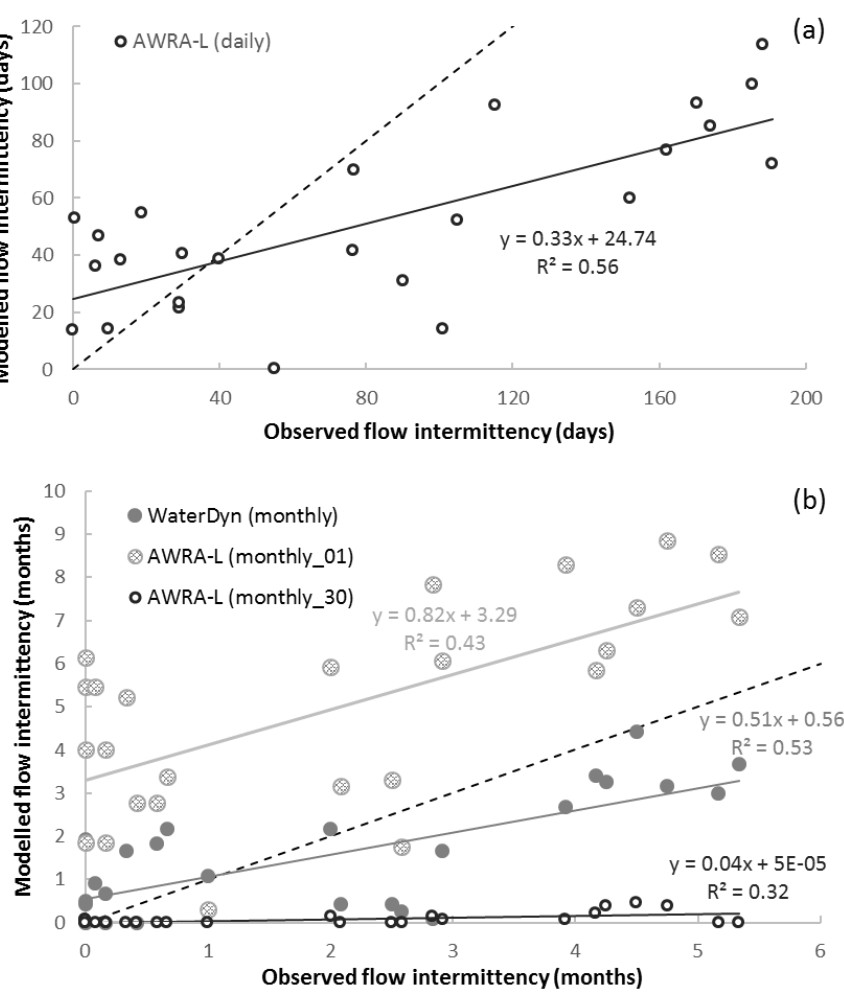


**Figure 8. Scatter plots of the observed and modelled flow intermittency by the two models (AWRA-L and WaterDyn model) for SEQ. daily AWRA-L and monthly WaterDyn are derived from the original data from the two models, while AWRA-L monthly_01 and monthly_30 are flow intermittency estimates using the two different aggregation methods with different thresholds (one day vs. 30 days) to classify a month as zero-flowing. The solid line represents the regression line for each model. The 1:1 line (dashed line) 665 is plotted for reference.**

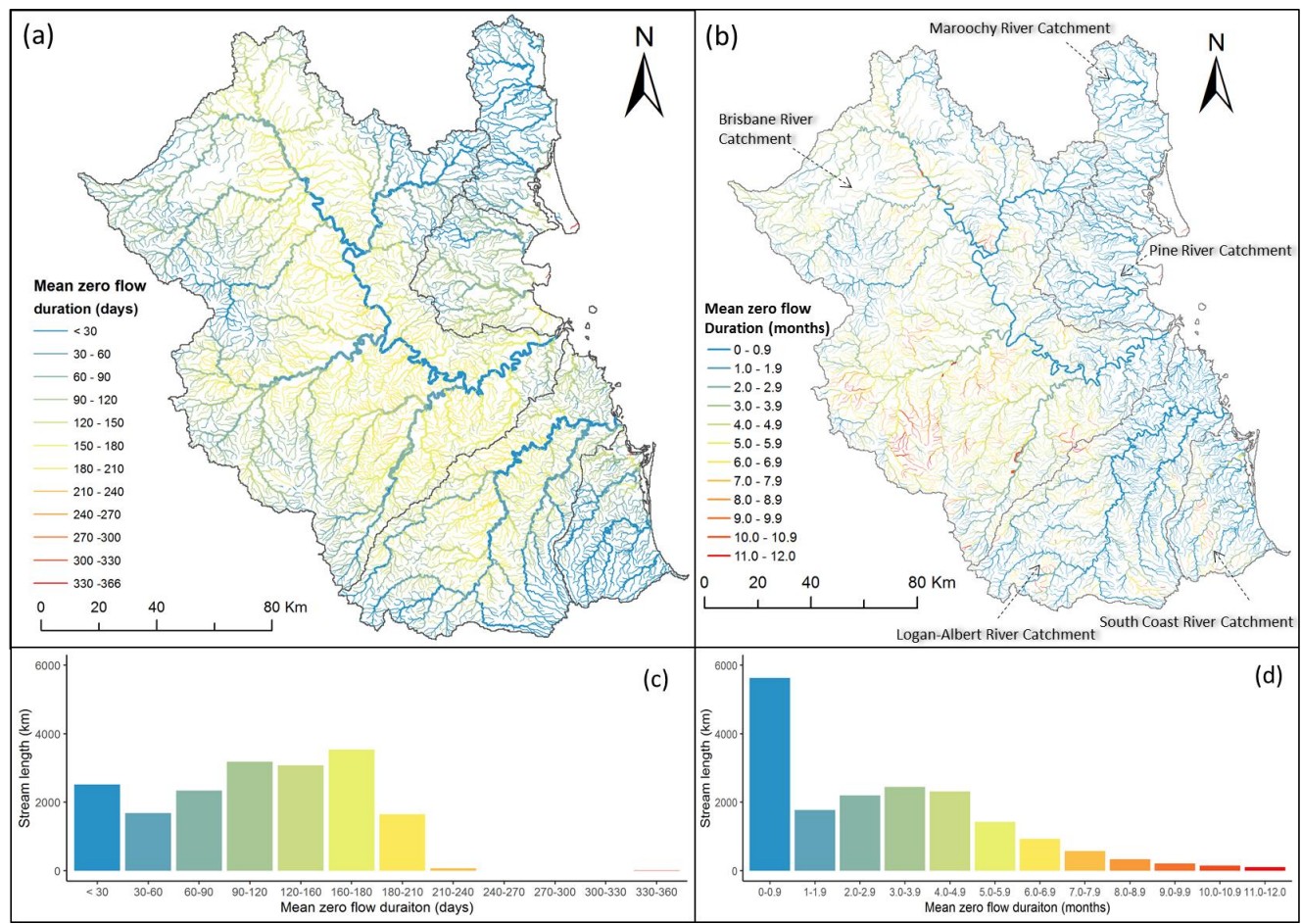

**Figure 9. Comparison of the spatial pattern of average annual flow intermittency in SEQ derived from (a) daily flow simulations from the AWRA-L model and (b) monthly flow simulations from the WaterDyn model. Stream segments in both figures are coloured using the same frame but different units. Line thicknesses show the stream orders. Frequency distributions of variations in the total stream length for each of twelve flow intermittency classes are also shown for (c) the AWRA-L model and (d) the WaterDyn model.**

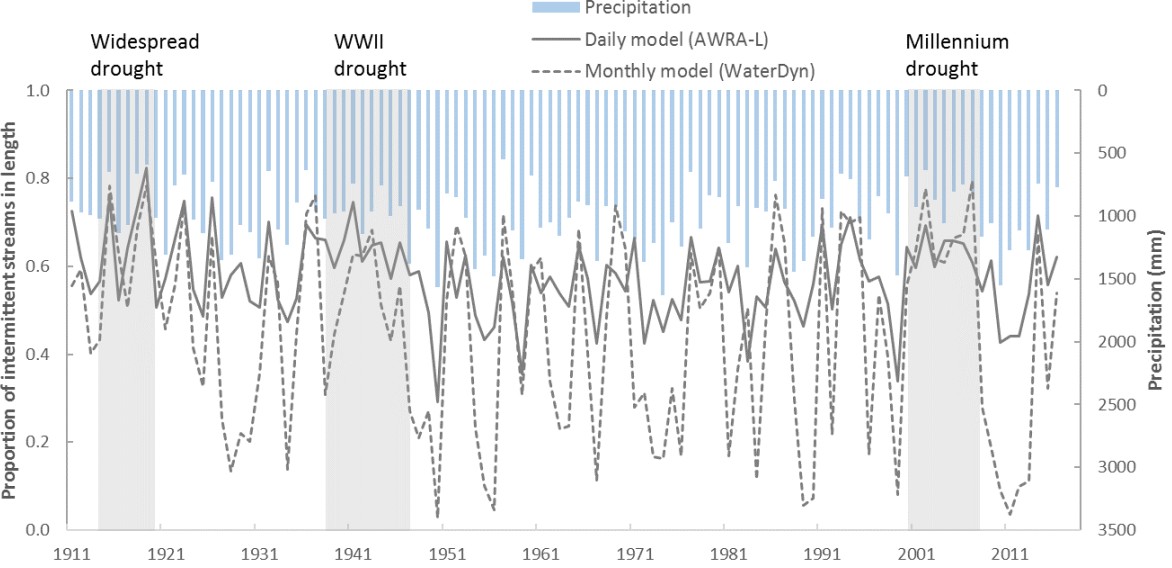

**Figure 10. Comparison of intra-annual variation of the proportion of intermittent streams in length from 1911-2016 across SEQ, derived from stream flow simulations from the daily flow model (lumped, solid grey line) and monthly flow model (grey dash line). Three severe droughts in Australia were also presented as transparent grey rectangles: Widespread drought (1914-1920), WWII droughts (1939-1946) and Millennium droughts (2001-2009). The time series of annual mean precipitation is shown for reference.**

**Table 1. Flow metrics used to describe average-, high- and low-flow conditions across key components of hydrological variation. Note that a spell independence criteria of 5 days was applied to regard periods between spells of less than 5 days as "in spell".**

| Conditions | Component | Abbreviation | Definition | Units |
|---|---|---|---|---|
| Average flow | Magnitude | Avg.magnitude | Mean daily flow for entire period | $m^3\ s^{-1}$ |
| | Variability | Avg.magnitude.cv | Coefficient of variation in mean daily flow | % |
| High-flow | Magnitude | H.magnitude | The average annual maximum flow | $m^3\ s^{-1}$ |
| | Timing | H.timing | The mean Julian date of annual maximum | unitless |
| | Variability | H.timing.cv | Coefficient of variation in Julian date of annual maximum flow | % |
| | Frequency | H.frequency | Mean of annual count of spells above the $90^{th}$ percentile flow | unitless |
| | Duration | H.duration | Mean duration of all spells above the $90^{th}$ percentile flow | days |
| | Rate of rise | H.rise | Mean rate of positive changes in flow from one day to the next | $m^3\ s^{-2}$ |
| | Rate of fall | H.fall | Mean rate of negative changes in flow from one day to the next | $m^3\ s^{-2}$ |
| Low-flow | Magnitude | L.magnitude | The average annual minimum flow | $m^3\ s^{-1}$ |
| | Timing | L.timing | The mean Julian date of annual minimum | unitless |
| | Variability | L.timing.cv | Coefficient of variation in Julian date of annual minimum flow | % |
| | Frequency | L.frequency | Mean of annual count of spells below the $10^{th}$ percentile flow | unitless |
| | Duration | L.duration | Mean duration of all spells below the $10^{th}$ percentile flow | days |