# Peer review of "Evaluating a landscape-scale daily water balance model to support spatially continuous representation of flow intermittency throughout stream networks"

_Hydrology and Earth System Sciences, 2020_

## Referee Comment (RC1) · Anonymous Referee #1 · 4 Mar 2020

Sunny Yu and coauthor present work modelling intermittent streamflow in two Australian catchments. Overall the paper is well-written and covers a topic that is of huge interest right now. I have some concerns about how the work is presented and will describe them below.

-Authors do not discuss how much data is required to assess low-flows. I fully acknowledge that it can be very difficult in getting long-term data sets. The data that they use here covers a period from January 1, 2005 through December 31, 2017, so 13 years. I do not think that this is a long enough period of time to quantify, characterize, or model

[Figure]

low-flows.

-Related to this, I am left wondering why the authors would choose a model that they know overestimates streamflow following a rainfall even when they are specifically interested in the lower-end of the streamflow continuum in systems that are dominated by streamflow that is mostly from rainfall.

-Some of the language is very vague. For instance on MS line 121-122 "If streamflow can be simulated at an acceptable accuracy...." but do not provide any guidelines for what is an acceptable accuracy.

-There is absolutely no explanation of the metrics that the authors used to characterize streamflow. Olden and Poff 2003 describe that it is really hard to characterize intermittent streams with metrics because the metrics that are used to describe one type of intermittent system are not the best to describe other types of intermittent systems. We also know that intermittent streams that are close in proximity to each other can behave very differently from each other from Margaret Zimmer, Adam Ward, and Katie Costigan's work. There's no discussion of metrics or how even close intermittent streams can behave differently in the manuscript. I also thought having to flip between a table and a figure to figure out what they were displaying could be improved.

-Related to this, the discussion seems to be over emphasizing the implications of the results. Yes, this is an important first step to modelling intermittent streamflow. However, their results only show that the modeled and observed streamflows have a r2 of less than 0.56! I would not say that this is "fair to good overall alignment" like the authors say in line 299. There is also no discussion, as I mentioned above, about how difficult it is to transfer metrics and results from this coastal Australian sites to other phyiographic areas. They admit that the two catchments used here are similar but the models preformed very differently for them.

I think that this is a great first step in being able to model intermittent streamflow. However, I am worried about the model choice overestimating low-flows– the ones

they focus on here– and I think that they are over emphasizing the strength of their findings. I think that we have a long way to go before we are able to apply the results of case studies like this to better understand intermittent systems as a whole, but this work is a great step towards that direction.

---

## Referee Comment (RC2) · Anonymous Referee #2 · 13 Mar 2020

The topic of this paper is certainly timely, and evaluation of how runoff routing, temporal resolution of models and climate impacts on spatial/temporal variability of drying streams is important. The biggest challenge I see in this manuscript is that the hypotheses presented in lines 163-166 are not clear and the framing of the problem and results wanders between computationally efficient streamflow routing to the timescale of importance, to the sub-catchment climate variability, to capturing spatial and temporal patterns of intermittency. With sufficient re-organization, additional details on the individual models and observational data, re-evaluation of the time period of no flow

that allows results to be compared across daily and monthly modes this work could provide interesting insights into intermittent stream research. Given the extent of revisions needed, I do not suggest it to be accepted at this time

Using out of the box hydrologic models (AWRA-L, AWAP) that over predict baseflow will certainly limited the ability to capture no-flow conditions (Figure 7, lines 309-311). These models are not fully described, even conceptually in the paper, making it challenging for a reader to understand which assumptions lead to this over-prediction. A previous study was used to benchmark flow intermittency, but was not explained in the methods.

Some of the methods of examining the low-flows themselves seem questionable, namely, that all days in a month had to have zero flow for the flows in that month to be zero from the AWRA-L outputs (line 187). There is work being done that suggests that a stream that goes dry for 15 days in a year is considered intermittent, so using a consistently dry 30 day window could be an exceptionally high threshold, either way, description of why a given threshold is used is necessary.

The timeframe of observational data included is not clear, and it was not presented with the comparison between model output in Figure 9. The modeled flow from 1911-2016 is included in the paper, with no reference to how well that model actually did at capturing low flows in the calibration time period. Figure 6 is slightly misleading because the dashed line is not a continuous variable and the catchment areas do not increase linearly.

The writing and organization of the manuscript could be improved throughout (e.g. 59 -63). References to the multiple model configurations throughout is particularly confusing (e.g. a table that has the 4 model configurations and associated details with acronyms would be useful). One important caveat relevant to modelling intermittent streams at a daily-time step using contiguous data is not referenced (e.g. stream gaging locations are generally put where there is usually surface water flow). There are

several quantitative results (r2) that are presented, yet the discussion poses that the models "showed fair to good overall alignment" which seems to overstate the ability to capture the low flows given how low the r2 was.

———————————————————

---

## Referee Comment (RC3) · George Allen (Referee) · 15 Mar 2020

The manuscript, "Evaluating a landscape-scale daily water balance model to support spatially continuous representation of flow intermittency throughout stream networks" by Yu et al. presents a study that quantifies the ability of two water balance models to simulate streamflow in two basins in Australia, with a particular focus on intermittent streamflow. The authors focus on comparing different water balance models with different timesteps and comparing a flow routing streamflow model to a simple lumped model. Perhaps the most novel component of the analysis involves the characterization of so-called cease-to-flow conditions by applying a gauge derived threshold to the streamflow simulations.

The manuscript is well written and of interest a variety of scientific communities including hydrologists, ecologists, and potentially biogeochemists. While there are a number of improvements that should be made to the manuscript (see below), I don't think there are any issues that warrant a major revision to this manuscript. The three most important issues that should be addressed are as follows:

1) The authors need to address the uncertainty of, and assumptions involved with, developing linear relationships at a limited number of gauges and then extrapolating these relationships across basins. For example, are the locations of the gauges a representative sample of the population of streams within the two basins, or are they biased towards large, perennial rivers segments? Another example: are the gauges used to calibrate the water-balance and streamflow models used in this study the same gauges used to estimate crease-to-flow occurrence? If so, please include how this fact may impact the results, particularly in terms of uncertainty.

2) When comparing monthly and daily models, the authors classify a month as no-flow only if every day of the month is estimated to be at zero flow. Wouldn't this approach bias the results to be more perennial? Is this why the daily model doesn't perform as well as the monthly model at the monthly timestep? Please provide some rationale on this decision for the monthly classification.

3) Finally, I suspect that Geofabric is missing some of the smallest streams (see Benstead & Leigh (2012) An expanded role for river networks, Nature Geoscience). If so, this error will control the proportion of rivers that are predicted to be intermittent, a primary finding of this study.

Specific comments:

Abstract:

L27: replace "intermittent flows" with "cease-to-flow events"

L28: add "at a monthly timestep" after "intermittency"

L29: add ", using a daily streamflow model" after "1911-2016". The monthly model produced a different estimate.

Main text:

L92: As mentioned above, add an acknowledgement about how the location of these reference gauges is likely biased towards particular river types (e.g. large perennial rivers) and river forms (e.g. narrow, single threaded rivers located near bridges), and how this bias might influence the extrapolation of the cease-to-flow threshold to all Geofabric stream segments.

L99: Add a little more information about Geofabric. What is the spatial resolution? Does it contain all of the smallest streams in the basins? If there is a channelization threshold, it will control the proportion of rivers that are estimated to be intermittent.

L112: As mentioned above, are the gauges used to calibrate the water-balance and streamflow models used in this study the same gauges used to estimate cease-to-flow? If so, please include how this fact may impact the results, particularly in terms of uncertainty.

L114: As mentioned above, please provide more information on the types of rivers and streams that these gauges are located on. This can help the reader understand the uncertainty associated with this analysis.

L122-123: "the readily available runoff data can be more accessible for potential applications" I don't follow this logic. Using a flow propagation model doesn't limit accessibly and should be relatively fast using RAPID, especially at the scale of these two catchments.

L160-161: "given that we do not have access to the underlying models to directly adjust

model parameters." RAPID is open source and you can adjust these parameters.

L162: Gauges are on rivers with large upstream drainage areas. There should be an acknowledgement that there are many smaller streams that likely have higher cease-to-flow occurrence and the gauges are likely not representative of these smaller streams.

L187: "all days in a month had to have zero flow for the flows for that month to be zero". Wouldn't this approach bias the results to be more perennial? Is this why the daily model doesn't perform as well as the monthly model at the monthly timestep? Please provide some rationale on this decision.

L197-198: As mentioned above, "The temporal pattern of flow intermittency was expressed as the proportion of streams with flow intermittency > 30 days or 1 month" – is this definition of intermittent streams based off of something or is it just arbitrary?

L239: insert "fair" before "match"

L288: Please explain "time of concentration" for the uninformed reader. Would be best to introduce it earlier on in the manuscript.

L300 and L301: typo: replace "KEG" with "KGE"

L318-319: "and recently many studies have developed methods to calculate transmission losses for better flow simulations (Lange, 2005; Costa et al., 2012)." The citations provided are neither recent nor many. L329: add "temporal" before "resolution"

L337: replace "is difference" with "are differences"

Figures:

Figure 3: Providing a y-axis with units would make it easier to interpret these boxplots

Figure 5: Perhaps considering scaling some of these y-axes as log, outliers make it difficult to compare the distributions and see the distribution of data where most of the data are located.

---

## Author Comment (AC1) · 17 Apr 2020

**Responses to Reviewer #1 comments on "Evaluating a landscape-scale daily water balance model to support spatially continuous representation of flow intermittency throughout stream networks" [hess-2020-10]**

We thank Reviewer #1 for providing these constructive comments that help improve the quality of this manuscript.

**Reviewer comment:**

Sunny Yu and coauthor present work modelling intermittent streamflow in two Australian catchments. Overall the paper is well-written and covers a topic that is of huge interest right now. I have some concerns about how the work is presented and will describe them below.

-Authors do not discuss how much data is required to assess low-flows. I fully acknowledge that it can be very difficult in getting long-term data sets. The data that they use here covers a period from January 1, 2005 through December 31, 2017, so 13 years. I do not think that this is a long enough period of time to quantify, characterize, or model low-flows.

**Authors reply:**

Thanks for the positive comments. The 13 year study period (01/01/2005-31/12/2017) is close to a discharge record length of 15 years which Kennard et al. (2010) concluded is sufficient to enable accurate estimation of low flow metrics. In addition, our study period begins in the middle of the Australian Millennium Drought (2001-2009), meaning that the low-flow assessment in this study included a significant low-flow period. Therefore, we believe that our study period is appropriate to assess low flows. We will add this justification in the revised manuscript.

**Reviewer comment:**

-Related to this, I am left wondering why the authors would choose a model that they know overestimates streamflow following a rainfall even when they are specifically interested in the lower-end of the streamflow continuum in systems that are dominated by streamflow that is mostly from rainfall.

**Authors reply:**

First, the AWRA-L model is able to provide spatially contiguous runoff outputs that are necessary to characterise flow intermittency throughout river networks. Equally important, the runoff outputs from the model are readily available and covers all of Australia from 1900 to the current year, enabling the potential application of the method proposed in this study to other areas.

In addition, due to the fundamental difference in runoff drivers for mean and peak flows and low flows, and that many hydrological models are calibrated to the average condition, low flows are usually overestimated (Smakhtin, 2001; Staudinger et al., 2011). To better apply the modelled low flows, we hypothesised and tested the potential causes of the AWRA-L model to overestimate low flows. Furthermore, to mitigate the overestimation of low flows by the model, we estimated segment-specific zero-flow thresholds by developing regression relationships relating gauged zero-flow duration to surrounding environmental variables.

It is also worth mentioning that the AWRA-L model used in this study has already been calibrated and validated for a range of hydrological conditions when it was being developed by CSIRO and the Australian Bureau of Meteorology (Viney et al., 2015). In this study, we applied more rigorous validations to the AWRA-L model, including evaluating its ability to estimate flow intermittency. To our knowledge, there was no other models that have undergone comparable calibration and validation processes for the Australian setting, making AWRA-L the most appropriate option for this study.

**Reviewer comment:**

-Some of the language is very vague. For instance on MS line 121-122 "If streamflow can be simulated at an acceptable accuracy...." but do not provide any guidelines for what is an acceptable accuracy.

**Authors reply:**

Thanks for pointing that sentence out. We will revise it to "If streamflow simulated with a routing model shows little difference to that without a routing model, then the conversion process can be more efficient …". We will also read through the manuscript thoroughly and revise all vague sentences to make their meaning more explicit.

**Reviewer comment:**

-There is absolutely no explanation of the metrics that the authors used to characterize streamflow. Olden and Poff 2003 describe that it is really hard to characterize intermittent streams with metrics because the metrics that are used to describe one type of intermittent system are not the best to describe other types of intermittent systems. We also know that intermittent streams that are close in proximity to each other can behave very differently from each other from Margaret Zimmer, Adam Ward, and Katie Costigan's work. There's no discussion of metrics or how even close intermittent streams can behave differently in the manuscript. I also thought having to flip between a table and a figure to figure out what they were displaying could be improved.

**Authors reply:**

We are assuming that the reviewer was referring to the flow metrics in Table 1. Those flow metrics have been commonly used to characterise flow regimes, but we will add more explanation of those metrics to the main text. However, we did not aim to use those metrics to characterise intermittent streams, but only to evaluate the ability of the AWRA-L model in simulating streamflow for different components (high-, average- and low-flows).

Based on the model discharge simulations, we further calculated the number of zero-flow days/months to characterise flow intermittency throughout river networks (described in Section 3.4), which is the main focus of this study. We agree with the reviewer that finding the most suitable metrics to characterise different types of intermittent streams is itself an important topic, however, it is beyond the scope of this research.

In our revision, we will revisit section 3.3 to ensure that the purpose of these flow metrics are presented explicitly. We will also add abbreviations of flow metrics shown in Figure 3 and 5 to Table 1, to make the figures and table better connected.

**Reviewer comment:**

-Related to this, the discussion seems to be over emphasizing the implications of the results. Yes, this is an important first step to modelling intermittent streamflow. However, their results only show that the modeled and observed streamflows have a r2 of less than 0.56! I would not say that this is "fair to good overall alignment" like the authors say in line 299. There is also no discussion, as I mentioned above, about how difficult it is to transfer metrics and results from this coastal Australian sites to other phyiographic areas. They admit that the two catchments used here are similar but the models preformed very differently for them.

**Authors reply:**

We feel that the reviewer may have misunderstood this part of model performance.

The $R^2$ value of 0.56 is not about the AWRA-L model performance in streamflow simulation, but about the modelled and observed flow *intermittency* (Figure 7), that latter being calculated from streamflow data as the number of days/months with zero flow. The metric used in this study to evaluate model performance in estimating daily streamflow is the Kling-Gupta efficiency (KGE; results presented in Section 4.2). KGE takes values from -1 to 1: KGE = 1 indicates perfect agreement between simulations and observations, and KGE < -0.41 indicates that the mean of observations provides better estimates than simulations. Our results showed that KGE values were more than -0.41 for all gauges, ranging from -0.19 to 0.76 in SEQ and 0.11 to 0.71 in the Tamar (lines 214-216). These results justify our interpretation that daily streamflow estimates showed a fair to good overall alignment with the observed flows in our study regions.

We evaluated model performance in two hydro-climatically distinctive regions of eastern Australia and found no significant difference in performance between regions (lines 216-218; 302). We also believe the proposed approach has a potential applicability to other regions of Australia and globally. All the data we used in this study to characterise flow intermittency are available for the Australian national scale, and similar datasets also exist in other countries. For example, similar to Geofabric used here, the National Hydrography Dataset Plus (NHDplus) and HydroSHEDS provide hydrographic dataset and hydro-environmental attributes for the national and global scale, respectively. In addition, similar to the daily flow model AWRA-L used in this study, other national-scale hydrologic models are also available around the world, such as the community Noah land surface model (Noah-MP) in the US and the HYPE model in Sweden. We will add more details to the Discussion regarding the applicability of our proposed approach.

References:

Kennard, M.J., Mackay, S.J., Pusey, B.J., Olden, J.D., Marsh, N., 2010. Quantifying uncertainty in estimation of hydrologic metrics for ecohydrological studies. River Research and Applications, 26(2): 137-156. DOI:10.1002/rra.1249

Smakhtin, V.U., 2001. Low flow hydrology: a review. Journal of Hydrology, 240(3): 147-186. DOI:https://doi.org/10.1016/S0022-1694(00)00340-1

Staudinger, M., Stahl, K., Seibert, J., Clark, M., Tallaksen, L., 2011. Comparison of hydrological model structures based on recession and low flow simulations. Hydrology and Earth System Sciences, 15(11): 3447-3459.

Viney, N. et al., 2015. AWRA-L v5.0: Technical description of model algorithms and inputs. CSIRO, Australia.

---

## Author Comment (AC2) · 17 Apr 2020

**Responses to Reviewer #2 comments on "Evaluating a landscape-scale daily water balance model to support spatially continuous representation of flow intermittency throughout stream networks" [hess-2020-10]**

We thank Reviewer #2 for providing these constructive comments that help improve the quality of this manuscript.

**Reviewer comment:**

The topic of this paper is certainly timely, and evaluation of how runoff routing, temporal resolution of models and climate impacts on spatial/temporal variability of drying streams is important. The biggest challenge I see in this manuscript is that the hypotheses presented in lines 163-166 are not clear and the framing of the problem and results wanders between computationally efficient streamflow routing to the timescale of importance, to the sub-catchment climate variability, to capturing spatial and temporal patterns of intermittency. With sufficient re-organization, additional details on the individual models and observational data, re-evaluation of the time period of no flow that allows results to be compared across daily and monthly modes this work could provide interesting insights into intermittent stream research. Given the extent of revisions needed, I do not suggest it to be accepted at this time.

**Authors reply:**

We thank the reviewer for pointing out several issues, however, we also believe that these issues can be sufficiently addressed in this revision.

First, we believe the framing of the research questions was clearly outlined in the last paragraph of the Introduction where we stated (lines 84-93):

"In this study, we sought to apply spatially contiguous daily runoff outputs from the AWRA-L water balance model to quantify the spatial extent and temporal patterns of flow intermittency. To assess the accuracy of the AWRA-L model for daily flow simulations, we first developed a simple but effective technique to convert runoff to streamflow for two hydro-climatically distinctive regions. …. We further assessed the uncertainty of the AWRA-L model in capturing patterns of flow intermittency. Lastly, we evaluated the effect of time step (daily vs. monthly) on the relative performance of the model in replicating observed patterns of cease to flow periods at reference gauges."

The hypotheses presented in lines 163-166 simply provided more details in the Methods section relating to the first objective (assessing the accuracy of the AWRA-L model for daily flow simulations).

We feel the structure and organisation of the paper was logically arranged to address our main objectives. Sections 4.1 and 4.2 of the Results evaluate the effects of streamflow routing and regional differences in hydro-climatic variability on streamflow simulations. Section 4.3 of the Results evaluated our ability to accurately estimate spatial and temporal patterns of flow intermittency using simulated spatially contiguous streamflow data. The importance of model time-step (daily vs. monthly) in estimation of streamflow intermittency was also evaluated in Section 4.3 of the Results.

We acknowledge that more details on the individual models and observational data should be provided, and we will do so in the revised manuscript.

Concerning the time period of no flow to enable comparison of daily and monthly models, we classified a month as no-flow only if every day of the month was estimated to be at zero flow. This classification method was aimed to convert daily flow intermittency to monthly flow intermittency, allowing the daily flow model AWRA-L to be comparable to the monthly flow model AWAP in terms of the ability to estimate flow intermittency. As the monthly flow model AWAP outputs monthly average flow, the zero value of monthly flow means all days in the month have zero flows. That's the reason why we chose that classification method. We will add the rationale in the revised manuscript. Additionally, we will also try using a different method to aggregate the modelled daily flow intermittency into monthly flow intermittency. This way, the results would be biased to be more "intermittent" as compared to our original results that may be biased to be more "perennial", and these two together should provide readers with both the upper and lower bounds of comparing daily and monthly models in estimating flow intermittency.

**Reviewer comment:**

Using out of the box hydrologic models (AWRA-L, AWAP) that over predict baseflow will certainly limited the ability to capture no-flow conditions (Figure 7, lines 309-311). These models are not fully described, even conceptually in the paper, making it challenging for a reader to understand which assumptions lead to this over-prediction. A previous study was used to benchmark flow intermittency, but was not explained in the methods.

**Authors reply:**

We will add more details on the individual models (including the AWAP model used to benchmark flow intermittency) and observational data as supplemental material to the manuscript.

We are not surprised that the AWRA-L model over-estimates low flows, a common problem with many hydrological models due to the difficulties of quantifying hydrological processes influencing low flow discharge (Smakhtin, 2001; Staudinger et al., 2011). In this study, we further investigated the potential sensitivity of the model to rainfall events by testing two hypothesis: 1) the overestimation of gains to low flow discharge, and 2) underestimation of transmission losses (Section 3.3; Figure 6). We also estimated appropriate zero-flow thresholds for each stream segment to mitigate this over-estimation of low flows.

**Reviewer comment:**

Some of the methods of examining the low-flows themselves seem questionable, namely, that all days in a month had to have zero flow for the flows in that month to be zero from the AWRA-L outputs (line 187). There is work being done that suggests that a stream that goes dry for 15 days in a year is considered intermittent, so using a consistently dry 30 day window could be an exceptionally high threshold, either way, description of why a given threshold is used is necessary.

**Authors reply:**

As explained in our response above, we classified a month as no-flow only if every day of the month was estimated to be at zero flow. This classification method was aimed to convert daily flow

intermittency to monthly flow intermittency, allowing the daily flow model AWRA-L to be comparable to the monthly flow model AWAP in terms of their ability to estimate flow intermittency. We will add this rationale in the revised manuscript. Additionally, we will also try using a different method to aggregate the modelled daily flow intermittency into monthly flow intermittency (in line with our first response).

**Reviewer comment:**

The timeframe of observational data included is not clear, and it was not presented with the comparison between model output in Figure 9. The modeled flow from 1911-2016 is included in the paper, with no reference to how well that model actually did at capturing low flows in the calibration time period. Figure 6 is slightly misleading because the dashed line is not a continuous variable and the catchment areas do not increase linearly.

**Authors reply:**

The timeframe of observational streamflow data included was clearly described in lines 111-112: "All gauges have less than 0.5 % missing values over the period from 01/01/2005 to 31/12/2017". This observational streamflow data was not presented in Figure 9; instead, the comparison between modelled and observed streamflow data is presented in Figure 4.

The AWRA-L model has already been calibrated and validated by its developers from the Australian Bureau of Meteorology and CSIRO (Viney et al., 2015). In our study, we further evaluated the model accuracy in streamflow simulations over the period of 2005-2017, with a particular focus on low flows. The model performance at capturing low flows was clearly illustrated in Figure 5. Based on the accuracy assessment, we applied a longer period (1911-2016) of the model outputs to estimate the temporal dynamics of flow intermittency in SEQ (Figure 9). We will provide more details about the calibration and validation of the AWRA-L during its development as supplemental material to the revised manuscript.

We agree with the Reviewer's comment regarding Figure 6. We will redraw this figure to remove the dashed line and only retain the value dots.

**Reviewer comment:**

The writing and organization of the manuscript could be improved throughout (e.g. 59-63). References to the multiple model configurations throughout is particularly confusing (e.g. a table that has the 4 model configurations and associated details with acronyms would be useful). One important caveat relevant to modelling intermittent streams at a daily-time step using contiguous data is not referenced (e.g. stream gaging locations are generally put where there is usually surface water flow). There are several quantitative results (r2) that are presented, yet the discussion poses that the models "showed fair to good overall alignment" which seems to overstate the ability to capture the low flows given how low the r2 was.

**Authors reply:**

We will thoroughly check the manuscript and improve the writing and clarity where necessary. However, as explained in an earlier response, we feel the organisation of the paper is logically arranged to address our main objectives. We will revise the sentence in lines 59-63 to: "These kind

of simulations are important to better understand the causes of flow intermittency at multiple spatial scales, and enable ecologically-relevant characterisation of streamflow properties such as the magnitude, frequency, duration, and rates of change in high or low flow events".

We will also include a table of all model configurations and other relevant information in the revised manuscript to improve clarity.

We disagree with the Reviewers' assertion that "stream gaging locations are generally put where there is usually surface water flow". In Australia this is not the case with around 70% of 830 streamflow gauges found to be located on streams with varying degrees of flow intermittency (Kennard et al., 2010). Therefore, the Reviewers' suggested caveat does not apply to the Australian situation.

The $R^2$ value of 0.56 does not relate to the AWRA-L model performance in streamflow simulation; instead, it relates to the concordance between modelled and observed flow _intermittency_ (see Figure 7), which was calculated from streamflow data as the number of days/months with zero flow. AWRA-L model performance was instead assessed using the Kling-Gupta efficiency (KGE) metric compare the alignment between modelled and observed streamflows. KGE takes values from -1 to 1: KGE = 1 indicates perfect agreement between simulations and observations, and KGE < -0.41 indicates that the mean of observations provides better estimates than simulations. Our results showed that KGE values were more than -0.41 for all gauges, ranging from -0.19 to 0.76 in SEQ and 0.11 to 0.71 in the Tamar (lines 214-216). These results justify our interpretation that daily streamflow estimates showed a fair to good overall alignment with the observed flows in our study regions.

In our revision, we will revisit the Methodology and Results sections to ensure that the evaluation procedures are explained more explicitly.

Reference:

Kennard, M.J. et al., 2010. Classification of natural flow regimes in Australia to support environmental flow management. Freshwater Biology, 55(1): 171-193. DOI:10.1111/j.1365-2427.2009.02307.x

Smakhtin, V.U., 2001. Low flow hydrology: a review. Journal of Hydrology, 240(3): 147-186. DOI:https://doi.org/10.1016/S0022-1694(00)00340-1

Staudinger, M., Stahl, K., Seibert, J., Clark, M., Tallaksen, L., 2011. Comparison of hydrological model structures based on recession and low flow simulations. Hydrology and Earth System Sciences, 15(11): 3447-3459.

Viney, N. et al., 2015. AWRA-L v5.0: Technical description of model algorithms and inputs. CSIRO, Australia.

---

## Author Response (AR1)

**Response to Editor Christa Kelleher's comments**

We thank the Editor Christa Kelleher for inviting us to revise the manuscript. We also thank her for summarising all Reviewers comments and providing constructive comments that help improve the quality of this manuscript.

**Editor comment:**

The manuscript has received three constructive reviews. All reviewers note that the topic of this manuscript is both important and timely and note that the study will be of interest to a broad readership.

Though the reviewers had several recommendations that I believe will be useful to the authors when revising the manuscript, there were two common themes throughout:

**Authors reply:**

We have addressed all of the editor and reviewer comments as detailed below.

**Editor comment:**

The reviewers all had several questions and comments regarding the methods section of this manuscript, and sought additional clarity. I encourage the authors to expand these sections with both text and equations, as they have outlined in their response, including a discussion of the model framework, the selected flow metrics (and why these were selected), and the choice of a metric of model accuracy (why KGE and not an explicitly low flow metric like LRMSE?). The reviewers also had questions regarding some of the decisions made by the authors in the methods section, and encouraged the authors to justify their decisions. In their response, the authors have developed a good plan to address these comments, and I'd encourage them to acknowledge not only their decisions and why they made them, but also to include a discussion of these limitations in the context of how to move the science of predicting intermittent streams forward.

**Authors reply:**

In response to reviewers comments related to the methods section, we have expanded the methodology section substantially to improve clarity. The major changes are:

1) We added details on the individual models (lines 121-125 for the AWRA-L model, and lines 220-224 for the WaterDyn (initially called "AWAP") model) and observational data (lines 114-118).
2) We provided more explanation of the selected flow metrics and the reason for selecting them (lines 150-154; 166-168). In addition, we also added abbreviations of flow metrics shown in Figure 4 and 6 to Table 1 to make the figures and Table better connected.
3) We included the KGE equation and associated descriptions. Regarding the choice of metric to evaluate model accuracy, we have provided further justification as follows: "The use of KGE provides an overall assessment of AWRA-L model performance, and the flow metrics in Table 1 are used to comprehensively evaluate the model accuracy for various components of flow regimes, including the flow metrics related to low flows" (lines 168-170). We therefore believe that the use of other low flow-focused metrics (e.g. LRMSE) is redundant as we have

already provided a detailed assessment of those related to low flows, including the magnitude, timing, duration, frequency of low flow spells (Table 1).

Regarding the method to classify a month as zero-flowing using daily streamflow, we have now added further rationale for the method we used in the original manuscript, namely that all days in a month had to have zero flow for the flows for that month to be zero. We have now included an additional method that considered a month as non-flowing when at least one day in the month had zero flow (lines 213-219). This way, the results would be biased to be more "intermittent" as compared to our original results that may be biased to be more "perennial". Consideration of results from both methods should provide readers with both the upper and lower bounds of comparing daily and monthly models in estimating flow intermittency. In addition, we discussed the implication of the two different methods in the revised manuscript (lines 377-381).

**Editor comment:**

The other common string of comments from the reviewers centered on some of the limitations of the model in the context of the physical hydrology of intermittent streams. For instance, as pointed out by the reviewers, the choice of flow metrics that best describe hydrologic regimes may differ across systems, and from a physical hydrologic perspective, even streams in close proximity can exhibit different behavior. I encourage the authors to acknowledge these limitations, and the implications of these limitations for their work, and for the work of the broader community working to simulate and study intermittent streams. Though the authors responded to these reviewer comments, I request that they revise their manuscript to provide greater discussion of these limitations in their revision.

**Authors reply:**

We agree that there are limitations of the model in runoff generation, particularly for low flow periods, and this has important implications for estimating spatio-temporal patterns in streamflow intermittency using modelled runoff data. To acknowledge this limitation, we have now included an expanded discussion of this issue (citing the work of Zimmer and colleague) in Section 5.2 of the Discussion as follows (Lines 353-357, new text highlighted in bold, with the original text provided for context):

"This suggests that the AWRA-L model is a generally robust model in predicting average- and high-flows, but still needs some improvement to better simulate low flows. **Runoff generation processes can vary substantially through space and time due to such factors as variations in soil depth, antecedent soil moisture and groundwater connectivity, and this can influence spatio-temporal variations in low flow characteristics, including streamflow intermittency (Zimmer and McGlynn, 2017). However, it is unknown the extent to which this contributed to uncertainty in the simulation of low flows and estimation in streamflow intermittency in this study**. The uncertainty of AWRA-L in low flow simulations can be linked to its over-responsiveness to rainfall, partly caused by overestimation of "in situ" gains and underestimation of transmission losses to low flow discharge, as shown in SEQ. Previous studies found that lateral flow exchange between grid cells of land surface models (e.g. AWRA-L) plays a significant role in redistributing soil water (Kim and Mohanty, 2016), and thus may improve "in situ" surface/subsurface runoff simulations (Lee and Choi, 2017). On the other hand, hydrological process involved in transmission losses have been extensively discussed (Jarihani et al., 2015; Konrad, 2006), and studies have developed methods to calculate transmission losses for better flow simulations (Costa et al., 2012; Lange, 2005). Therefore,

low flow simulations by AWRA-L can possibly be improved by incorporating lateral flow exchange algorithms and better accounting for hydrological process such as evapotranspiration from riparian vegetation and infiltration into channel beds. This improvement is made more likely as AWRA-L has been released as a Community Modelling System (https://github.com/awracms/awra_cms), which allows co-development by the research community."

**Editor comment:**

One reviewer also recommended changes to figures, to make them more interpretable, and I encourage the authors to incorporate these recommendations into their revision.

**Authors reply:**

We have incorporated these recommendations into the revision as detailed below:

1) We have now provided unit for each of the flow metrics in Figure 4 and 6 (Figure 3 and 5 in the original submission).
2) We have scaled the y axes as $\log_{10}$ in Figure 6 to better display the distribution of the data.
3) We have re-drawn Figure 7 as a scatter plot to improve clarity.

**Editor comment:**

I request that the authors incorporate their proposed revisions into the discussion paper for re-review.

**Authors reply:**

We have incorporated all of our proposed revisions into the revised manuscript for re-review.

**Responses to Reviewer #1 comments on "Evaluating a landscape-scale daily water balance model to support spatially continuous representation of flow intermittency throughout stream networks" [hess-2020-10]**

We thank Reviewer #1 for providing these constructive comments that help improve the quality of this manuscript.

**Reviewer comment:**

Sunny Yu and coauthor present work modelling intermittent streamflow in two Australian catchments. Overall the paper is well-written and covers a topic that is of huge interest right now. I have some concerns about how the work is presented and will describe them below.

-Authors do not discuss how much data is required to assess low-flows. I fully acknowledge that it can be very difficult in getting long-term data sets. The data that they use here covers a period from January 1, 2005 through December 31, 2017, so 13 years. I do not think that this is a long enough period of time to quantify, characterize, or model low-flows.

**Authors reply:**

Thanks for the positive comments. The 13 year study period (01/01/2005-31/12/2017) is close to a discharge record length of 15 years which Kennard et al. (2010) concluded is sufficient to enable accurate estimation of low flow metrics. In addition, our study period begins in the middle of the Australian Millennium Drought (2001-2009), meaning that the low-flow assessment in this study included a significant low-flow period. Therefore, we believe that our study period is appropriate to assess low flows. It is also worth noting that our study period is the common period covered by streamflow observations across all gauges, which provides a consistent assessment of model performances spatially. We have added this justification in the revised manuscript (lines 181-184).

**Reviewer comment:**

-Related to this, I am left wondering why the authors would choose a model that they know overestimates streamflow following a rainfall even when they are specifically interested in the lower-end of the streamflow continuum in systems that are dominated by streamflow that is mostly from rainfall.

**Authors reply:**

First, the AWRA-L model is able to provide spatially contiguous runoff outputs that are necessary to characterise flow intermittency throughout river networks. Equally important, the runoff outputs from the model are readily available and covers all of Australia from 1900 to the current year, enabling the potential application of the method proposed in this study to other areas.

In addition, due to the fundamental difference in runoff drivers for mean and peak flows and low flows, and that many hydrological models are calibrated to the average condition, low flows are usually overestimated (Smakhtin, 2001; Staudinger et al., 2011). To better apply the modelled low flows, we hypothesised and tested the potential causes of the AWRA-L model to overestimate low flows. Furthermore, to mitigate the overestimation of low flows by the model, we estimated segment-specific zero-flow thresholds by developing regression relationships relating gauged zero-flow duration to surrounding environmental variables.

It is also worth mentioning that the AWRA-L model used in this study has already been calibrated and validated for a range of hydrological conditions when it was being developed by CSIRO and the Australian Bureau of Meteorology (Viney et al., 2015). In this study, we applied more rigorous validations to the AWRA-L model, including evaluating its ability to estimate flow intermittency. To our knowledge, there was no other models that have undergone comparable calibration and validation processes for the Australian setting, making AWRA-L the most appropriate option for this study.

**Reviewer comment:**

-Some of the language is very vague. For instance on MS line 121-122 "If streamflow can be simulated at an acceptable accuracy...." but do not provide any guidelines for what is an acceptable accuracy.

**Authors reply:**

Thanks for pointing that sentence out. We have revised it to "If streamflow simulated with a routing model shows little difference to that without a routing model, then the conversion process can be more efficient …" (lines 130-132). We also rephrased all vague sentences we could find to make their meaning more explicit. For example, one vague sentence once read:

"Given that we do not have access to the underlying model to directly adjust model parameters, we instead compared the observed and modelled low flow magnitude…"This has now been more specific about "the underlying model". The revised sentence now reads:

"Given that we do not have access to the AWRA-L model to directly adjust model parameters, we instead compared the observed and modelled low flow magnitude…"

Two additional examples are given below:

Original sentence:

"Due to the fact that water balance models often over-predict the magnitude of very low flows, we adopted the same method used in Yu et al (2018)…"

Revised sentence (lines: 198-200):

"Given the fact that water balance models often over-predict the magnitude of very low flows due to the difficulties of quantifying hydrological processes influencing low flow discharge, we adopted the same method used in Yu et al (2018)…"

Original sentence:

"The spatial patterns of flow intermittency derived from the daily and monthly flow simulations aligned well only for the main stems and some coastal streams…"

Revised sentence (lines: 287-288):

"The spatial patterns of flow intermittency derived from the daily AWRA-L and monthly WaterDyn flow simulations aligned well only for the main stems and some coastal streams…"

**Reviewer comment:**

-There is absolutely no explanation of the metrics that the authors used to characterize streamflow. Olden and Poff 2003 describe that it is really hard to characterize intermittent streams with metrics because the metrics that are used to describe one type of intermittent system are not the best to describe other types of intermittent systems. We also know that intermittent streams that are close in proximity to each other can behave very differently from each other from Margaret Zimmer, Adam Ward, and Katie Costigan's work. There's no discussion of metrics or how even close intermittent streams can behave differently in the manuscript. I also thought having to flip between a table and a figure to figure out what they were displaying could be improved.

**Authors reply:**

The metrics used to characterise streamflow are defined in Table 1. These flow metrics have commonly been used to characterise critical components of flow regimes across average, high, and low flow conditions. In the revised manuscript, we have added more explanation of these metrics with the following text (lines 150-152):

"The calculated flow metrics are commonly used to describe the critical components of flow regimes across average, high, and low flow conditions, including flow magnitude and variability, the timing, frequency and duration of high and low flows, and rates of changes in flow events (Olden and Poff, 2003; Poff et al., 1997)."

We have also added abbreviations of flow metrics shown in Figure 4 and 6 to Table 1, to make the figures and table better connected.

To quantify flow intermittency throughout river networks, we calculated the total number of zero-flow days per year based on the spatially contiguous modelled streamflow simulations. We also evaluated the effect of time step (daily vs. monthly) on the relative performance of the model in replicating observed patterns of cease to flow periods by comparison with flow intermittency estimates derived from a monthly model (Yu et al., 2018).

We agree with the Reviewer's important point that "intermittent streams that are close in proximity to each other can behave very differently from each other" as this has important implications for estimating spatio-temporal patterns in streamflow intermittency using modelled runoff data. To address this issue, we have now included an expanded discussion of this issue (citing the work of Zimmer and colleague) in Section 5.2 of the Discussion as follows (Lines 353-357, new text highlighted in bold, with the original text provided for context):

"This suggests that the AWRA-L model is a generally robust model in predicting average- and high-flows, but still needs some improvement to better simulate low flows. **Runoff generation processes can vary substantially through space and time due to such factors as variations in soil depth, antecedent soil moisture and groundwater connectivity, and this can influence spatio-temporal variations in low flow characteristics, including streamflow intermittency (Zimmer and McGlynn, 2017). However, it is unknown the extent to which this contributed to uncertainty in the simulation of low flows and estimation in streamflow intermittency in this study**. The uncertainty of AWRA-L in low flow simulations can be linked to its over-responsiveness to rainfall, partly caused by overestimation of "in situ" gains and underestimation of transmission losses to low flow discharge, as shown in SEQ. Previous studies found that lateral flow exchange between grid cells of land surface models (e.g. AWRA-L) plays a significant role in redistributing soil water (Kim and

Mohanty, 2016), and thus may improve "in situ" surface/subsurface runoff simulations (Lee and Choi, 2017). On the other hand, hydrological process involved in transmission losses have been extensively discussed (Jarihani et al., 2015; Konrad, 2006), and studies have developed methods to calculate transmission losses for better flow simulations (Costa et al., 2012; Lange, 2005). Therefore, low flow simulations by AWRA-L can possibly be improved by incorporating lateral flow exchange algorithms and better accounting for hydrological process such as evapotranspiration from riparian vegetation and infiltration into channel beds. This improvement is made more likely as AWRA-L has been released as a Community Modelling System (https://github.com/awracms/awra_cms), which allows co-development by the research community."

**Reviewer comment:**

-Related to this, the discussion seems to be over emphasizing the implications of the results. Yes, this is an important first step to modelling intermittent streamflow. However, their results only show that the modeled and observed streamflows have a r2 of less than 0.56! I would not say that this is "fair to good overall alignment" like the authors say in line 299. There is also no discussion, as I mentioned above, about how difficult it is to transfer metrics and results from this coastal Australian sites to other phyiographic areas. They admit that the two catchments used here are similar but the models preformed very differently for them.

**Authors reply:**

We feel that the reviewer may have misunderstood this aspect of our assessment of model performance.

The assessment of "fair to good overall alignment" related to the AWRA-L model performance in streamflow simulation. The metric used in this study to evaluate model performance in estimating daily streamflow is the Kling-Gupta efficiency (KGE; results presented in Section 4.2). KGE takes values from -1 to 1: KGE = 1 indicates perfect agreement between simulations and observations, and KGE < -0.41 indicates that the mean of observations provides better estimates than simulations. Our results showed that KGE values were more than -0.41 for all gauges, ranging from -0.19 to 0.76 in SEQ and 0.11 to 0.71 in the Tamar (lines 248-250). These results justify our interpretation that daily streamflow estimates showed a fair to good overall alignment with the observed flows in our study regions.

In contrast, the $R^2$ value of 0.56 relates to the assessment of concordance between the modelled and observed flow *intermittency* (Figure 8), which was calculated from streamflow data as the number of days/months with zero flow. In the revised manuscript, we have reworded aspects of the Methodology and Results sections to ensure that the evaluation procedures are explained more clearly. In addition, in the Discussion (Section 5.2), we now also provide an expanded discussion of the potential sources of uncertainty in the ability of the AWRA-L model to accurately simulate low flows (as detailed in the response above).

We evaluated model performance in two hydro-climatically distinctive regions of eastern Australia and found no significant difference in performance between regions (lines 250-252). We also believe the proposed approach has a potential applicability to other regions of Australia and globally. All the data we used in this study to characterise flow intermittency are available for the Australian national scale, and similar datasets also exist in other countries. We have added this information to the Discussion regarding the applicability of our proposed approach (lines 393-400):

[revised manuscript text omitted]

We thank Reviewer #2 for providing these constructive comments that help improve the quality of this manuscript.

**Reviewer comment:**

The topic of this paper is certainly timely, and evaluation of how runoff routing, temporal resolution of models and climate impacts on spatial/temporal variability of drying streams is important. The biggest challenge I see in this manuscript is that the hypotheses presented in lines 163-166 are not clear and the framing of the problem and results wanders between computationally efficient streamflow routing to the timescale of importance, to the sub-catchment climate variability, to capturing spatial and temporal patterns of intermittency. With sufficient re-organization, additional details on the individual models and observational data, re-evaluation of the time period of no flow that allows results to be compared across daily and monthly modes this work could provide interesting insights into intermittent stream research. Given the extent of revisions needed, I do not suggest it to be accepted at this time.

**Authors reply:**

We thank the reviewer for pointing out several issues, which we believe can be readily addressed in this revision.

First, we believe the framing of the research questions was clearly outlined in the last paragraph of the Introduction where we stated (lines 85-94):

"In this study, we sought to apply spatially contiguous daily runoff outputs from the AWRA-L water balance model to quantify the spatial extent and temporal patterns of flow intermittency. To assess the accuracy of the AWRA-L model for daily flow simulations, we first developed a simple but effective technique to convert runoff to streamflow for two hydro-climatically distinctive regions. ….
We further assessed the uncertainty of the AWRA-L model in capturing patterns of flow intermittency. Lastly, we evaluated the effect of time step (daily vs. monthly) on the relative performance of the model in replicating observed patterns of cease to flow periods at reference gauges."

The hypotheses presented in lines 192-195 (lines 163-166 in the original manuscript) simply provided more details in the Methods section relating to the first objective (assessing the accuracy of the AWRA-L model for daily flow simulations).

We feel the structure and organisation of the paper was logically arranged to address our main objectives. Sections 4.1 and 4.2 of the Results evaluate the effects of streamflow routing and regional differences in hydro-climatic variability on streamflow simulations. Section 4.3 of the Results evaluated our ability to accurately estimate spatial and temporal patterns of flow intermittency using simulated spatially contiguous streamflow data. The importance of model time-step (daily vs. monthly) in estimation of streamflow intermittency was also evaluated in Section 4.3 of the Results.

We have now provided more details on the individual models (lines 121-125 for AWRA-L and lines 220-224 for WaterDyn (initially called "AWAP") and observational data (lines 114-118) in the revised manuscript.

Concerning the time period of no flow to enable comparison of daily and monthly models, we classified a month as no-flow only if every day of the month was estimated to be at zero flow. This classification method was aimed to convert daily flow intermittency to monthly flow intermittency, allowing the daily flow model AWRA-L to be comparable to the monthly flow model WaterDyn in terms of the ability to estimate flow intermittency. As the monthly flow model WaterDyn outputs monthly average flow, the zero value of monthly flow means all days in the month have zero flows. Additionally, we have now evaluated an alternative method to aggregate the modelled daily flow intermittency into monthly flow intermittency (lines 213-219). We assessed the effect of considering a month as non-flowing when at least one day in the month had zero flow. This way, the results would be biased to be more "intermittent" as compared to our original results that may be biased to be more "perennial", and these two together should provide readers with both the upper and lower bounds of comparing daily and monthly models in estimating flow intermittency. This is expanded upon in response to a detailed Reviewer comment on this issue below.

**Reviewer comment:**

Using out of the box hydrologic models (AWRA-L, AWAP) that over predict baseflow will certainly limited the ability to capture no-flow conditions (Figure 7, lines 309-311). These models are not fully described, even conceptually in the paper, making it challenging for a reader to understand which assumptions lead to this over-prediction. A previous study was used to benchmark flow intermittency, but was not explained in the methods.

**Authors reply:**

We have now added more details on the individual models (including the AWRA-L model to simulate daily flows and the WaterDyn (initially called "AWAP") model used to benchmark flow intermittency based on monthly flow simulations) to the manuscript (lines 121-125 for AWRA-L and lines 220-224 for WaterDyn).

We are not surprised that the AWRA-L model over-estimated low flows, a common problem with many hydrological models due to the difficulties of quantifying hydrological processes influencing low flow discharge (Smakhtin, 2001; Staudinger et al., 2011). In this study, we further investigated the potential sensitivity of the model to rainfall events by testing two hypothesis: 1) the overestimation of gains to low flow discharge, and 2) underestimation of transmission losses (Section 3.3; Figure 7). We also estimated appropriate zero-flow thresholds for each stream segment to mitigate this over-estimation of low flows. In the Discussion (Section 5.2), we also provide an expanded discussion of the potential sources of uncertainty in the ability of the AWRA-L model to accurately simulate low flows (as per the response to this issue raised by Reviewer #1)

**Reviewer comment:**

Some of the methods of examining the low-flows themselves seem questionable, namely, that all days in a month had to have zero flow for the flows in that month to be zero from the AWRA-L outputs (line 187). There is work being done that suggests that a stream that goes dry for 15 days in

a year is considered intermittent, so using a consistently dry 30 day window could be an exceptionally high threshold, either way, description of why a given threshold is used is necessary.

**Authors reply:**

As explained in our response above, we classified a month as no-flow only if every day of the month was estimated to be at zero flow. This classification method was aimed to convert daily flow intermittency to monthly flow intermittency, allowing the daily flow model AWRA-L to be comparable to the monthly flow model WaterDyn in terms of their ability to estimate flow intermittency. We have added this rationale in the revised manuscript. Additionally, we also added a different method to aggregate the modelled daily flow intermittency into monthly flow intermittency (in line with our first response).

**Reviewer comment:**

The timeframe of observational data included is not clear, and it was not presented with the comparison between model output in Figure 9. The modeled flow from 1911-2016 is included in the paper, with no reference to how well that model actually did at capturing low flows in the calibration time period. Figure 6 is slightly misleading because the dashed line is not a continuous variable and the catchment areas do not increase linearly.

**Authors reply:**

The timeframe of observational streamflow data included was clearly described in lines 113-114: "All gauges have less than 0.5 % missing values over the period from 01/01/2005 to 31/12/2017". This observational streamflow data was not presented in Figure 10 (Figure 9 in the original submission); instead, the comparison between modelled and observed streamflow data is presented in Figure 5.

The AWRA-L model has already been calibrated and validated by its developers from the Australian Bureau of Meteorology and CSIRO (Viney et al., 2015). In our study, we further evaluated the model accuracy in streamflow simulations over the period of 2005-2017, with a particular focus on low flows. The model performance at capturing low flows was clearly illustrated in Figure 6. Based on the accuracy assessment, we applied a longer period (1911-2016) of the model outputs to estimate the temporal dynamics of flow intermittency in SEQ (Figure 10). We have provided more details about the calibration and validation of the AWRA-L during its development in the revised manuscript (lines 124-125, 170-173).

We agree with the Reviewer's comment regarding Figure 6. We have re-drawn this figure as a scatter plot to improve clarity (see Figure 7).

**Reviewer comment:**

The writing and organization of the manuscript could be improved throughout (e.g. 59-63). References to the multiple model configurations throughout is particularly confusing (e.g. a table that has the 4 model configurations and associated details with acronyms would be useful). One important caveat relevant to modelling intermittent streams at a daily-time step using contiguous data is not referenced (e.g. stream gaging locations are generally put where there is usually surface water flow). There are several quantitative results (r2) that are presented, yet the discussion poses

that the models "showed fair to good overall alignment" which seems to overstate the ability to capture the low flows given how low the r2 was.

**Authors reply:**

We have thoroughly checked the manuscript and improved the writing and clarity where necessary.

The sentence on line 59-63 has been reworded as follows:

"These kinds of simulations are important to better understand the causes of flow intermittency at multiple spatial scales and enable ecologically-relevant characterisation of streamflow properties such as the magnitude, frequency, duration, and rate of change in high or low flow events". (lines 60-63)

Other examples of improvements to writing and clarity are provided in our responses to comments by Reviewer #1.

Regarding organization of the manuscript, we feel the paper is logically arranged to address our main objectives (as explained in an earlier response). To improve clarity of the various model configurations used in the paper, we have now included a new figure (instead of a table as the Reviewer suggested) of all model configurations and other relevant information in the revised manuscript (see Figure 2).

We disagree with the Reviewers' assertion that "stream gaging locations are generally put where there is usually surface water flow". In Australia this is not the case with around 70% of 830 streamflow gauges found to be located on streams with varying degrees of flow intermittency (Kennard et al., 2010). Therefore, the Reviewers' suggested caveat does not apply to the Australian situation.

Regarding the comment about our assessment of the ability to accurately model low flows, we addressed this issue in response to comments by Reviewer # 1.

The assessment of "fair to good overall alignment" related to the AWRA-L model performance in streamflow simulation. The metric used in this study to evaluate model performance in estimating daily streamflow is the Kling-Gupta efficiency (KGE; results presented in Section 4.2). KGE takes values from -1 to 1: KGE = 1 indicates perfect agreement between simulations and observations, and KGE < -0.41 indicates that the mean of observations provides better estimates than simulations. Our results showed that KGE values were more than -0.41 for all gauges, ranging from -0.19 to 0.76 in SEQ and 0.11 to 0.71 in the Tamar (lines 248-250). These results justify our interpretation that daily streamflow estimates showed a fair to good overall alignment with the observed flows in our study regions.

In contrast, the $R^2$ value of 0.56 relates to the assessment of concordance between the modelled and observed flow *intermittency* (Figure 8), which was calculated from streamflow data as the number of days/months with zero flow.

In the revised manuscript, we have reworded aspects of the Methodology and Results sections to ensure that the evaluation procedures are explained more clearly. In addition, in the Discussion (Section 5.2), we now also provide an expanded discussion of the potential sources of uncertainty in the ability of the AWRA-L model to accurately simulate low flows (see lines 353-357).

We thank George Allen for providing these constructive comments that help improve the quality of this manuscript.

**Reviewer comment:**

The manuscript, "Evaluating a landscape-scale daily water balance model to support spatially continuous representation of flow intermittency throughout stream networks" by Yu et al. presents a study that quantifies the ability of two water balance models to simulate streamflow in two basins in Australia, with a particular focus on intermittent streamflow. The authors focus on comparing different water balance models with different timesteps and comparing a flow routing streamflow model to a simple lumped model. Perhaps the most novel component of the analysis involves the characterization of so-called cease-to-flow conditions by applying a gauge derived threshold to the streamflow simulations.

The manuscript is well written and of interest a variety of scientific communities including hydrologists, ecologists, and potentially biogeochemists. While there are a number of improvements that should be made to the manuscript (see below), I don't think there are any issues that warrant a major revision to this manuscript. The three most important issues that should be addressed are as follows:

**Authors reply:**

Thanks for the positive comments.

**Reviewer comment:**

1) The authors need to address the uncertainty of, and assumptions involved with, developing linear relationships at a limited number of gauges and then extrapolating these relationships across basins. For example, are the locations of the gauges a representative sample of the population of streams within the two basins, or are they biased towards large, perennial rivers segments? Another example: are the gauges used to calibrate the water-balance and streamflow models used in this study the same gauges used to estimate crease-to-flow occurrence? If so, please include how this fact may impact the results, particularly in terms of uncertainty.

**Authors reply:**

We agree that the spatial distribution of gauged streams as a representative sample of the population of streams is an important consideration when calibrating a regression model and using it to extrapolate more widely. We considered our sampled gauge locations to be representative of the population of streams and included an updated statement to this effect in the revised manuscript lines 114-118: "The gauges were widely dispersed throughout each study area and encompassed a range of stream sizes, catchment areas (22-3,881 km$^2$ in SEQ; 33-3,294 km$^2$ in Tamar) and flow regime types, ranging from highly intermittent to perennial streams (see results). Therefore, we regard the selected gauges to be representative of the environmental and hydrological conditions in

the regions, except for extremely small catchments with an area < 22 km$^2$ that likely have higher cease-to-flow occurrence."

We here describe in more detail the sets of streamflow gauges used in the various steps in our analyses and illustrate this in the following Figure R1. The water balance model (AWRA-L) was both calibrated and validated by the developers from the Australian Bureau of Meteorology and CSIRO at the national scale (Viney et al., 2015), with 301 gauges used for calibration and a different set of 304 gauges used for validation (Zhang et al., 2013). Our study converted the AWRA-L water balance model predictions to streamflow estimates and these were validated for different components of the flow regime (high-, average- and low flows), using 25 and 15 gauges in two hydro-climatically distinctive regions, respectively (SEQ and Tamar). Only 6 of the 25 gauges in SEQ and 3 of the 15 gauges in Tamar were the same as those used to calibrate the AWRA-L water balance model. This small overlap between the AWRA-L calibration gauge set (n=301) and the streamflow model validation gauge set (n=25 in SEQ and 15 in Tamar) means that potential overestimation of streamflow model performance is likely to be minimal. We have now included the following new text in the revised manuscript (lines 170-173) to reflect this:

"Only six of the 25 gauges in SEQ and three of the 15 gauges in Tamar were the same as those used to calibrate the AWRA-L water balance model. This small overlap between the AWRA-L calibration gauge set (n=301) and the streamflow model validation gauge set (n=25 in SEQ and 15 in Tamar) means that potential overestimation of streamflow model performance is likely to be minimal."

A larger set of 43 gauges in SEQ (including 21 of the 25 gauges used by us for streamflow validation) was used to estimate the zero-flow threshold for this region. However, because the validation of the streamflow model applied to the raw discharge simulations, rather than the corrected discharge simulations with zero-flow thresholds, we do not regard this choice of streamflow gauges to be an issue for model validation in this study.

[Figure]

Figure R1. Schematic illustration of the streamflow gauge sets used in the different modelling processes described in this paper.

**Reviewer comment:**

2) When comparing monthly and daily models, the authors classify a month as no-flow only if every day of the month is estimated to be at zero flow. Wouldn't this approach bias the results to be more perennial? Is this why the daily model doesn't perform as well as the monthly model at the monthly timestep? Please provide some rationale on this decision for the monthly classification.

**Authors reply:**

As detailed in our responses to similar comments by Reviewer # 2, we classified a month as no-flow only if every day of the month was estimated to be at zero flow. This classification method was aimed to convert daily flow intermittency to monthly flow intermittency, allowing the daily flow model AWRA-L to be comparable to the monthly flow model WaterDyn (initially called "AWAP") in terms of the ability to estimate flow intermittency. As the monthly flow model WaterDyn outputs monthly average flow, the zero value of monthly flow means all days in the month have zero flows. Additionally, we have now evaluated an alternative method to aggregate the modelled daily flow intermittency into monthly flow intermittency (lines 213-219). We assessed the effect of considering a month as non-flowing when at least one day in the month had zero flow. This way, the results would be biased to be more "intermittent" as compared to our original results that may be biased to be more "perennial", and these two together should provide readers with both the upper and lower bounds of comparing daily and monthly models in estimating flow intermittency.

**Reviewer comment:**

3) Finally, I suspect that Geofabric is missing some of the smallest streams (see Benstead & Leigh (2012) An expanded role for river networks, Nature Geoscience). If so, this error will control the proportion of rivers that are predicted to be intermittent, a primary finding of this study.

**Authors reply:**

We agree that small streams comprise a large proportion of river networks, may be more frequently intermittent than larger streams, and their prevalence may be underestimated using readily available spatial datasets such as used in our study. We do not, however, regard these issues as compromising the main objectives of our study.

We believe that the spatial resolution of the smallest streams identified in the Geofabric stream network (version 2.1.1) is appropriate considering the relatively large spatial extent of our study areas. The Geofabric stream network is a fully connected and directed stream network derived from the national 9 arc-second DEM and flow direction grid (~250m resolution). Streams of seven Strahler orders were delineated in Geofabric for the study river networks, with the minimum upstream drainage area of 1.5 km$^2$, while the two study areas (SEQ and Tamar) are 21,331 km$^2$ and 11,215 km$^2$, respectively. In addition, the Geofabric is the finest resolution national stream network layer with supporting environmental attributes available for Australia and is of much finer resolution than similar products such as HydroSHEDS (15 arc-second (~500 m) resolution).

Moreover, an updated version of Geofabric (version 3) is now being developed (http://www.bom.gov.au/water/geofabric/about.shtml). The new version is based on a finer scale digital elevation model (~30m resolution) and aims to provide continent-wide river networks with eight Strahler stream orders. Our proposed approach to characterising flow intermittency can also be built upon this new version of Geofabric.

**Specific comments:**

Abstract:

L27: replace "intermittent flows" with "cease-to-flow events"

L28: add "at a monthly timestep" after "intermittency"

L29: add ", using a daily streamflow model" after "1911-2016". The monthly model produced a different estimate.

Main text:

L92: As mentioned above, add an acknowledgement about how the location of these reference gauges is likely biased towards particular river types (e.g. large perennial rivers) and river forms (e.g. narrow, single threaded rivers located near bridges), and how this bias might influence the extrapolation of the cease-to-flow threshold to all Geofabric stream segments.

L99: Add a little more information about Geofabric. What is the spatial resolution? Does it contain all of the smallest streams in the basins? If there is a channelization threshold, it will control the proportion of rivers that are estimated to be intermittent.

L112: As mentioned above, are the gauges used to calibrate the water-balance and streamflow models used in this study the same gauges used to estimate cease-to-flow? If so, please include how this fact may impact the results, particularly in terms of uncertainty.

L114: As mentioned above, please provide more information on the types of rivers and streams that these gauges are located on. This can help the reader understand the uncertainty associated with this analysis.

L122-123: "the readily available runoff data can be more accessible for potential applications" I don't follow this logic. Using a flow propagation model doesn't limit accessibly and should be relatively fast using RAPID, especially at the scale of these two catchments.

L160-161: "given that we do not have access to the underlying models to directly adjust model parameters." RAPID is open source and you can adjust these parameters.

L162: Gauges are on rivers with large upstream drainage areas. There should be an acknowledgement that there are many smaller streams that likely have higher cease-to-flow occurrence and the gauges are likely not representative of these smaller streams.

L187: "all days in a month had to have zero flow for the flows for that month to be zero". Wouldn't this approach bias the results to be more perennial? Is this why the daily model doesn't perform as well as the monthly model at the monthly timestep? Please provide some rationale on this decision.

L197-198: As mentioned above, "The temporal pattern of flow intermittency was expressed as the proportion of streams with flow intermittency > 30 days or 1 month" – is this definition of intermittent streams based off of something or is it just arbitrary?

L239: insert "fair" before "match"

L288: Please explain "time of concentration" for the uninformed reader. Would be best to introduce it earlier on in the manuscript.

L300 and L301: typo: replace "KEG" with "KGE"

L318-319: "and recently many studies have developed methods to calculate transmission losses for better flow simulations (Lange, 2005; Costa et al., 2012)." The citations provided are neither recent nor many. L329: add "temporal" before "resolution"

L337: replace "is difference" with "are differences"

Figures:

Figure 3: Providing a y-axis with units would make it easier to interpret these boxplots

Figure 5: Perhaps considering scaling some of these y-axes as log, outliers make it difficult to compare the distributions and see the distribution of data where most of the data are located.

**Authors reply:**

All of the above comments have been addressed as suggested, except for the following, for which we provide individual responses below:

**Specific comment:**

L92: As mentioned above, add an acknowledgement about how the location of these reference gauges is likely biased towards particular river types (e.g. large perennial rivers) and river forms (e.g. narrow, single threaded rivers located near bridges), and how this bias might influence the extrapolation of the cease-to-flow threshold to all Geofabric stream segments.

**Authors reply:**

As explained in our response to the first major comment (above), we considered our sampled gauge locations to be representative of the population of streams and included a statement about this in the manuscript (lines 114-118): "The gauges were widely dispersed throughout each study area and encompassed a range of stream sizes, catchment areas (22-3,881 km$^2$ in SEQ; 33-3,294 km$^2$ in Tamar) and flow regime types, ranging from highly intermittent to perennial streams (see results). Therefore, we regard the selected gauges to be representative of the environmental and hydrological conditions in the regions, except for extremely small catchments with an area < 22 km$^2$ that likely have higher cease-to-flow occurrence."

**Specific comment:**

L122-123: "the readily available runoff data can be more accessible for potential applications" I don't follow this logic. Using a flow propagation model doesn't limit accessibly and should be relatively fast using RAPID, especially at the scale of these two catchments.

**Authors reply:**

This sentence is in the context of whether the conversion process can be more efficient without a routing model. Here we actually mean that if the conversion process does not need a routing model (e.g. RAPID), the users of the AWRA-L runoff data can confidently skip the routing process, which makes the runoff data more accessible for potential applications.

**Specific comment:**

L160-161: "given that we do not have access to the underlying models to directly adjust model parameters." RAPID is open source and you can adjust these parameters.

**Authors reply:**

Here "the underlying models" was meant to be the AWRA-L model. We have revised the sentence as "given that we do not have access to the AWRA-L model to directly adjust model parameters."

**Specific comment:**

L187: "all days in a month had to have zero flow for the flows for that month to be zero". Wouldn't this approach bias the results to be more perennial? Is this why the daily model doesn't perform as well as the monthly model at the monthly timestep? Please provide some rationale on this decision.

**Authors reply:**

This comment is same to the second major comment. Please refer to our response to that major comment on Page 3.

**Specific comment:**

L197-198: As mentioned above, "The temporal pattern of flow intermittency was expressed as the proportion of streams with flow intermittency > 30 days or 1 month" – is this definition of intermittent streams based off of something or is it just arbitrary?

**Authors reply:**

This thresholds of cease-to-flow duration were chosen to make the flow intermittency estimates from daily and monthly models comparable, assuming 1 month = 30 days.

**Specific comment:**

Figure 3: Providing a y-axis with units would make it easier to interpret these boxplots

**Authors reply:**

We have now provided unit for each of the flow metrics in Figure 4 and 6 (Figure 3 and 5 in the original submission).

References:

[revised manuscript text omitted]

---

## Author Response (AR2)

**Response to Editor Christa Kelleher's comments**

We thank the Editor Christa Kelleher for summarising all Reviewers comments and providing constructive comments that help improve the quality of this manuscript.

**Editor comment:**

The article has received two additional reviews, both of which indicate that the manuscript is strong and has improved in the recent round of iteration.

**Authors reply:**

Thanks for the positive comments.

**Editor comment:**

However, both reviewers still had concerns. Reviewer 1 notes that their concerns were not adequately addressed, and encourages some additional text be added to the manuscript. Reviewer 2 would like additional information, particularly with regards to the threshold for intermittence (though other concerns are noted in the review report and should also be addressed) and recommends an additional figure that shows spatial variability in intermittence.

In particular, I agree with Reviewer 2 that additional details on the regression approach (including approach and outcomes) should be added, even though these methods have already been published in another article. All readers need enough information to ensure they fully understand the methods used in this particular study. As this is a key step for determining intermittency, more information is needed.

**Authors reply:**

We have responded in detail to each of the Reviewers comments and provide a brief summary here.

To address Reviewer 1's further comments, we have now added further details on the representativeness of stream gauges (Section 3.1) and acknowledged that the distribution of selected stream gauges and other factors may be a source of uncertainty for the application of the regression model (Section 3.4).

We have also added a new sentence in the revised Discussion (lines 405-408) to acknowledge that Geofabric may not be able to resolve the smallest streams in the study area, and thus our analyses may have under-estimated the proportion of intermittent streams.

Regarding Reviewer 2's comments, we have now added more details in Section 3.4 describing the linear regression models, including the model fits, methods for model evaluation, assessment of spatial autocorrelation and sources of uncertainty.

We have also added to Figure 9 two new inset plots showing frequency distributions of total stream length by flow intermittency classes identified from AWRA-L daily and WaterDyn monthly streamflow simulations, respectively. These revisions to Figure 9 now amply demonstrate the variations in flow intermittency across the study region.

**Editor comment:**

Both reviewers also note that the implications of this work, and limitations of this approach, should be further discussed.

**Authors reply:**

As indicated in our reply to the first comment, we have now acknowledged potential limitations and sources of uncertainty in our study.

We have also restructured a paragraph in the discussion Section 5.3 to make one of our main findings more apparent (new text highlighted in bold, with the original text provided for context).

"**Our results suggest that the temporal resolution of analysis should be dictated by the resolution of input streamflow data. More specifically**, the daily AWRA-L flow showed promise for estimating flow intermittency at a daily time step, while the monthly WaterDyn model was better than the monthly AWRA-L model in flow intermittency estimation at a monthly time step. This suggests that monthly flow models can sometimes outperform daily flow models in quantifying flow intermittency, depending on the intended temporal resolution of the analysis. For example, daily flow models may be appropriate for studies aimed at quantifying ecological responses to short term flow events, while monthly flow models are more suitable for research requiring the average degree of flow intermittency at a large spatial or temporal scale, such as examining the effect of flow intermittency on aquatic/streamside vegetation or species distributions (Stromberg et al., 2005). In addition, our study also suggested that the suitability of a monthly model (WaterDyn) for monthly resolution of analysis was not challenged by a daily model (AWRA-L) simply through aggregating daily streamflow simulations to a monthly time step. The aggregation methods used here applied one day or 30 days as a threshold and, respectively, either substantially overestimated or underestimated flow intermittency."

**Editor comment:**

I encourage the authors to incorporate these recommendations for adding some clarity to the manuscript.

**Authors reply:**

We have incorporated all these recommendations in the revised manuscript. Please also see our detailed responses to each of the Reviewer comments.

**Responses to George Allen's comments on "Evaluating a landscape-scale daily water balance model to support spatially continuous representation of flow intermittency throughout stream networks" [hess-2020-10]**

We thank George Allen for providing further comments that help improve the quality of this manuscript.

**Reviewer comment:**

While the authors have addressed almost all of my concerns with the initial version of the manuscript, I view that two of the major issues that I brought up have still not been adequately addressed in the text:

The first concern I raised previously was that I thought the placement of the stream gauges might be biased and not representative of the true hydrological conditions of the entire stream networks. In response, the authors state that the stream gauges are distributed across a range of streams types and therefore are a representative sample. However, they do not provide any evidence that the gauges are actually representative. Instead of asserting that the stream gauges are representative, I would suggest just acknowledging in the text that they are assuming that the gauges are representative and are a source of uncertainty in the analysis.

**Authors reply:**

We have now added further details on the representativeness of stream gauges and acknowledged that the distribution of selected stream gauges and other factors may be a source of uncertainty for the application of the regression model. The new text (highlighted in bold) reads:

"The gauges were widely dispersed throughout each study area and encompassed a range of stream sizes, catchment areas ($22 - 3{,}881$ km$^2$ in SEQ; $33 - 3{,}294$ km$^2$ in Tamar) and flow regime types, ranging from highly intermittent to perennial streams (see results). **However, the set of stream gauges used in our analyses under-represented the frequency of small low-order streams in both regions.** Therefore, we regard the selected gauges to be representative of the range of environmental and hydrological conditions in the regions, except for extremely small catchments with an area < 22 km$^2$ that likely have higher cease-to-flow occurrence." (lines 118-123)

"The unexplained variation in the predictive model **may be due to the limited number of environmental attribute covariates used in the model and hence ability to adequately represent the range of environmental processes that influence streamflow intermittency. Additional uncertainty in model predictions may arise because the distribution of stream gauges used for model calibration under-represented the frequency of extremely small catchments that likely had higher cease-to-flow occurrence**." (lines 227-230)

**Reviewer comment:**

The second concern is that they largely sidestep my concern that Geofabric doesn't contain data for the smallest streams in the study basin. Yes, I understand that Geofabric is probably the best dataset that is available to conduct this analysis but it still is not able to resolve the finest streams (e.g., see Shanafield et al., 2020, JGR-ES, https://doi.org/10.1029/2019JF005330 or Allen et al., 2018, Nature Communications, https://doi.org/10.1038/s41467-018-02991-w). I suggest that the authors

acknowledge in the text that Geofrabric is not able to resolve the smallest streams in the study area, and therefore the study's analysis likely is biased toward perennial streams when predicting the proportion of streams that are predicted to be intermittent.

In my view, adding these caveats and uncertainties to the text will only strengthen the manuscript and the study will remain interesting and useful to the scientific community.

**Authors reply:**

We apologise that the reviewer felt we "largely sidestepped" their concern about the spatial resolution of the Geofabric – we attempted to directly respond to their concern and further elaborate here.

We believe that the spatial resolution of the smallest streams identified in the Geofabric stream network (version 2.1.1) is appropriate to achieve the aims of our study (i.e. to apply spatially contiguous daily runoff outputs from the AWRA-L water balance model to quantify the spatial extent and temporal patterns of flow intermittency), considering the grain size of the streamline mapping data relative to the large spatial extent of our study areas. We agree that stream mapping at a finer grain size would no doubt reveal the existence of smaller streams, however, the key question is whether spatial patterns of flow intermittency would differ substantially depending on the spatial resolution of the data used to define these patterns. While such an assessment is beyond the scope of this study, the key issues relevant to our paper are the requirement that fine-grained streamline mapping data is actually available, is spatially consistent and is spatially comparable to evaluate spatio-temporal dynamics of streamflow intermittency (the primary objective of this paper). The Geofabric stream network dataset meets all of these requirements.

Nonetheless, we have now added a new sentence in the revised Discussion to acknowledge that Geofabric may not be able to resolve the smallest streams in the study area, and thus our analyses may have under-estimated the proportion of intermittent streams (Lines 405-408):

"However, given the limited spatial resolution of the Geofabric stream network data (9 arc-second longitude-latitude resolution, with the minimum upstream drainage area of 1.5 km$^2$) and hence ability to resolve the smallest streams, and that small streams are more likely to be intermittent, the proportion of predicted intermittent streams in SEQ may be under-estimated in our study."

**Responses to Anonymous Referee #4's comments on "Evaluating a landscape-scale daily water balance model to support spatially continuous representation of flow intermittency throughout stream networks" [hess-2020-10]**

We thank Anonymous Referee #4 for providing these constructive comments that help improve the quality of this manuscript.

**Reviewer comment:**

Below, I provide a short overview of the paper, my recommendation, and then more specific comments.

Broad overview. This paper addresses a critical problem for water resource management, regulatory, and research communities – how do we quantify the spatial variability in intermittence across broad spatial scales? Authors address this by applying a water balance model across two watersheds in Australia: Southeast Queensland (temperate to subtropical climates) and Tamar River of Tasmania (temperate climate). The authors modeling approach can be divided into three steps: (i) developing the AWRA-L runoff simulations, (ii) comparing AWRA-L runoff simulations with and without routing, and (iii) quantifying flow intermittency using daily and monthly outputs from the AWRA-L runoff model and comparing those outputs to the WaterDyn model. The authors highlight their model fit using flow regime metrics across multiple gages in each region, highlight that [atleast in this exercise] a routing model is was not necessary to simulate streamflow, and finally demonstrate spatial and temporal trends in intermittency across the two regions.

Recommendation. My recommendation is to accept with minor revisions. I am reviewing this paper for the first time, but it is in its second round of revision. This paper provides a unique solution to estimating large scale spatial and temporal patterns in intermittency. Further, authors should be commended for quantifying the fit [or in some cases lack thereof] between the simulated and observed streamflow regime. My one major concern is the method used to define the threshold for intermittence. I hope the authors will consider providing more information to their readers. Below, I provide several comments in an attempt to help authors improve an already good manuscript.

**Authors reply:**

Thanks for the positive comments. To address the major concern, we have now added more information for the method used to define the threshold for streamflow intermittency. For details, see our responses to the following two comments.

**Reviewer comment:**

Specific comments.

-At L205, authors outline their process of defining 'thresholds' for intermittence in the simulated flow data. I encourage the authors to provide information about the distributions of the flow quantiles of these thresholds (either in text, or better yet, a figure!).

**Authors reply:**

We have now provided the ranges and the median value of these thresholds for streamflow intermittency in the revised manuscript (lines 225-226): "The adopted thresholds ranged from 0 – 1.668 m$^3$/s, with a median value of 0.002 m$^3$/s."

**Reviewer comment:**

-Related to the last comment, authors need to provide more information about the linear models. What were the model fits like, were a split sample design employed, and did the authors consider spatial autocorrelation? The reader currently cannot adequately evaluate this portion of the analysis, which is a critical step when defining intermittency.

**Authors reply:**

We have now added more details in Section 3.4 describing the linear regression models (lines 205-231), including the model fits, methods for model evaluation, assessment of spatial autocorrelation and sources of uncertainty. The section now reads:

"We used linear regression to model the mean annual zero flow duration (daily time step) at each gauge as a function of catchment environment variables. This regression analysis was only conducted in SEQ as most gauges in the Tamar catchment had perennial flow. The environmental variables were the same as those in Yu et al. (2018), and included variables related to climate (annual daily maximum temperature), catchment geology topography (catchment area, catchment average slope, and catchment average elevation), and catchment soil properties (catchment average saturated hydraulic conductivity). Regression models were developed using all possible predictor variable combinations and we selected the "best" model for predicting zero flow duration based on corrected Akaike's Information Criterion (AICc) (Hurvich and Tsai, 1989). To estimate the prediction error of the selected model, we applied leave-one-out cross validation on the selected 43 gauges and reported prediction error ($R^2$) to estimate the model prediction performance. Regression model development and cross-validation were conducted with the MuMIn and boot packages in R (R Development, 2012). Regression analyses were performed on all combinations of predictor variables and the best model with the lowest AICc (-54.2) retained five covariates, including annual daily maximum temperature, catchment area, slope, average elevation, and average saturated hydraulic conductivity. The developed predictive model showed a good model fit with an adjusted $R^2$ of 0.71, and the leave-one-out cross validation on the regression model showed relatively good model performance with an average $R^2$ of 0.64. We checked for spatial autocorrelation of the regression model residuals (as recommended by Dormann et al., 2007) and found they were not significantly autocorrelated (Moran's I = -0.06, *p* = 0.69). Examination of spatial residual maps further supported this conclusion, with no spatial trends in model residuals apparent.

Next, we used the predictive models to extrapolate estimates of overall flow intermittency (in terms of the proportion of days with zero flow) to each segment throughout the river network. Finally, for each segment, the time-series of daily runoff was truncated (flows below the threshold were set to "0") by adopting an appropriate threshold of "zero flow" that preserved the proportion of days with flow as estimated in the previous step. The adopted thresholds ranged from 0 – 1.668 m$^3$/s, with a median value of 0.002 m$^3$/s. We recognise several sources of uncertainty in our approach to estimating the zero flow thresholds. The unexplained variation in the predictive model may be due to the limited number of environmental attribute covariates used in the model and hence ability to adequately represent the range of environmental processes that influence streamflow intermittency. Additional uncertainty in model predictions may arise because the distribution of

stream gauges used for model calibration under-represented the frequency of extremely small catchments that likely had higher cease-to-flow occurrence."

**Reviewer comment:**

-It's unclear why authors use parametric comparative statistics (i.e., t-test). I would suggest switching to a nonparametric analogs (i.e., Wilcoxon test) or showing normality to prove t-test is appropriate.

**Authors reply:**

Although the majority of variables show normality, measured by the Shapiro-Wilk Normality Test, we agree that a nonparametric Wilcoxon test is more appropriate. We have now replaced Student's *t*-test with Wilcoxon rank sum test in the revised manuscript (see new Figure 4), and the new results only change for one variable (i.e. the duration of low flow spells), whose *p* value changes from 0.08 to 0.03. These analyses do not change the main finding that the routing algorithm had negligible effects on flow simulations in our study areas.

**Reviewer comment:**

- After my first read-through, I was unclear why the WaterDyn model was incorporated into the manuscript. While this is reported in L220, I would strongly recommend authors make this more apparent [i.e., not hidden at the bottom of a paragraph.]

**Authors reply:**

We have improved clarity of this paragraph by bringing upfront the reason to incorporate the WaterDyn model to make it more apparent. The new added sentence reads: "To evaluate the effect of time step (daily vs. monthly) on the relative performance of the AWRA-L model in replicating observed patterns of cease-to-flow periods, we compared model outputs with those derived from a monthly water balance model – the WaterDyn model (Fig. 2). Monthly flow intermittency estimated from WaterDyn was thus used to benchmark results from the monthly AWRA-L." (lines 233-236)

**Reviewer comment:**

-Related to the comment above, there is a lot of important information packed in the paragraph starting at 370. [Infact, I would go so far as to say this is one of the main findings of the paper!] My suggestion is to rewrite this paragraph; lead with the argument that "our results suggest that the temporal resolution of analysis should be dictated by the resolution of input streamflow data;" and potentially further develop this discussion point. This is a useful result – one that will help others build on this studies results! Authors might even consider adding this result to the abstract.

**Authors reply:**

We agree that this finding should be more strongly emphasised. We have now brought the main point from the bottom in the paragraph to the beginning. The paragraph now starts with "Our results suggest that the temporal resolution of analysis should be dictated by the resolution of input

streamflow data". We also made minor text changes in the paragraph, while the majority remained unchanged, as they well fit with the new structure.

In addition, this key finding is highlighted in the abstract (lines 28-29): "The daily flow model did not perform better than the monthly flow model in quantifying flow intermittency at a monthly time step, and model selection should depend on the intended application of the model outputs".

**Reviewer comment:**

-Finally, I was disappointed the authors did not do more to explore the spatial variation of intermittence in their data. I encourage them to add an additional figure to highlight spatial variability. Note, I'm not suggesting they add an additional map. More appropriately, I would encourage authors to add a figure that quantifies spatial variability in intermittence.

**Authors reply:**

We agree that adding an additional figure to demonstrate spatial variability in flow intermittency is necessary and important. We have now added to Figure 9 two new inset plots showing frequency distributions of total stream length by flow intermittency classes identified from AWRA-L daily and WaterDyn monthly streamflow simulations, respectively. These revisions to Figure 9 now amply demonstrate the variations in flow intermittency across the study region.

[revised manuscript text omitted]